# Spatiotemporal dynamics of homologous recombination repair at single collapsed replication forks

Donna R. Whelan [1,2], Wei Ting C. Lee [1], Yandong Yin [1], Dylan M. Ofri[1], Keria Bermudez-Hernandez[1], Sarah Keegan[1], David Fenyo[1] & Eli Rothenberg[1]

Homologous recombination (HR) is a crucial pathway for the repair of DNA double-strand breaks. BRCA1/2 breast cancer proteins are key players in HR via their mediation of RAD51 nucleofilament formation and function; however, their individual roles and crosstalk in vivo are unknown. Here we use super-resolution (SR) imaging to map the spatiotemporal kinetics of HR proteins, revealing the interdependent relationships that govern the dynamic interplay and progression of repair events. We show that initial single-stranded DNA/RAD51 nucleofilament formation is mediated by RAD52 or, in the absence of RAD52, by BRCA2. In contrast, only BRCA2 can orchestrate later RAD51 recombinase activity during homology search and resolution. Furthermore, we establish that upstream BRCA1 activity is critical for BRCA2 function. Our analyses reveal the underlying epistatic landscape of RAD51 functional dependence on RAD52, BRCA1, and BRCA2 during HR and explain the phenotypic similarity of diseases associated with mutations in these proteins.

[1] Department of Biochemistry and Molecular Pharmacology, Perlmutter Cancer Center, New York University School of Medicine, New York, NY 10016, USA. [2] Department of Pharmacy and Applied Science, La Trobe Institute for Molecular Science, La Trobe University, Bendigo, VIC, Australia. Correspondence and requests for materials should be addressed to E.R. (email: Eli.Rothenberg@nyumc.org)

Double-strand breaks (DSBs) are an unavoidable consequence of daily replicative and transcriptional stress in all dividing cells. When left unrepaired or misrepaired, these breaks can lead to mutagenesis or cell death[1]. Given the necessity of high-fidelity repair, several complementary repair pathways have evolved that together constitute a holistic DNA damage response (DDR) signaling cascade involving a multitude of proteins. Two principal repair mechanisms have been identified and characterized: a relatively fast and somewhat lower-fidelity non-homologous end joining (NHEJ) pathway, and a slower, but more accurate, homologous recombination (HR) pathway[2]. While HR is preferred because of its use of a homologous strand as a template to avoid errors, it should not occur during G1 because of the absence of a suitable homologous sequence. Similarly, NHEJ cannot be used to repair single-ended DSBs (seDSBs) because of its requirement for two blunt DNA ends. Because the collapse of replication forks (RFs) has been shown to be the main source of endogenous DSBs, with lesions (including those caused by endogenous processes involving single-strand break induction) ahead of the replicon resulting in characteristic seDSBs, it is understood that HR is the dominant repair pathway for endogenous breaks[3–7].

Many proteins have been identified as contributors to the endogenous HR pathway, with a range of proposed functional interactions between these proteins and the damaged DNA[3,8]. MRE11-mediated resection at DSBs generates single-stranded DNA (ssDNA), committing the break to HR repair (HRR)[9,10]. This ssDNA is immediately coated with RPA, which is later replaced with RAD51 recombinase to form the ssDNA/RAD51 nucleofilament responsible for orchestrating homology search and strand invasion[3]. Once a homologous sequence is identified, it is thought that DNA polymerases synthesize DNA to replace any missing genetic information prior to either rescue of the collapsed RF or ligation to DNA synthesized by a converging fork, thus completing repair[11].

It has been established that the breast cancer susceptibility proteins BRCA1 and BRCA2 have critical roles in HR; homozygous knockout of either of these proteins is embryonically lethal in mice[12,13]. In humans, harmful mutations in either of the corresponding genes correlates with an increased risk of breast, ovarian, pancreatic, and prostate cancers[14,15]. Moreover, it has been shown that such mutations, as well as protein depletion, cause sensitivity to DSB-inducing drugs and increased replication stress[16,17]. Currently, the HR-related role of BRCA1 in vivo is ill-defined[15]. While there is evidence that it functions upstream of BRCA2[18], BRCA1 has also been implicated in DDR signaling, checkpoint activation, resection mediation, and recruitment of other proteins[18,19]. In contrast, BRCA2 is understood to have a single principal action: to act in mediating the ssDNA/RAD51 interaction necessary for homology search and recombination[8,16,20,21]. However, the mechanism by which BRCA2 facilitates ssDNA/RAD51 function and the impact of BRCA1 deficiencies on BRCA2 are unknown[8], an issue confounded by a lack of consensus regarding the intricacies of BRCA2's role as a mediator in RAD51 function[22,23].

The similarity of mutant BRCA1 and BRCA2 disease phenotypes presumably reflects a degree of functional overlap between the two proteins[21]. This potential crosstalk is highlighted by the recent surprising discovery of synthetic lethality in cells deficient in RAD52 and any one of BRCA1, BRCA2, PALB2 (a protein considered to function as a scaffold for BRCA1/BRCA2 interactions), or RAD51 paralogs[24–27]. This is of particular interest because of the absence of any disease phenotype associated with mutations in RAD52, despite the colocalization of RAD52 and RAD51 at damage foci, indicating some role for RAD52 in HR[28]. The epistatic relationships between these RAD51 mediators and the potential for redundant interactions or pathways are thus major unanswered questions in establishing the mechanism of HR[29].

A particular difficulty in defining the spatiotemporal progression of HR in vivo has been the limitations on spatial resolution and sensitivity conferred by conventional fluorescence microscopy. Owing to the diffraction of light, details of foci and colocalization within fluorescently labeled cellular samples are inherently limited and involve uncertainties spanning hundreds of nanometers. This makes it impossible to distinguish between clustered and individual proteins or damage foci. Furthermore, because of the high and homogeneous intracellular protein concentrations, successful detection of a fluorescent focus routinely requires the presence of dozens of fluorophores[30], a condition that in turn requires the induction of potent, clustered damage. Here we overcome these limitations by using multicolor super-resolution (SR) imaging, which provides a ten-fold improvement in spatial resolution as well as single-molecule sensitivity[31,32]. In combination with assays that specifically label nascent DNA (naDNA), ssDNA, and proteins associated with repair foci, we could detect and examine individual seDSB sites in vivo, routinely detecting several hundred individual naDNA and protein foci in a single-cell image with tens to hundreds of overlaps. This allowed the elucidation of the specific spatiotemporal features of HRR, including the previously undefined individual roles, crosstalk, and epistasis between BRCA1, BRCA2, RAD51, and RAD52.

## Results

**Super-resolution imaging for spatiotemporal mapping of repair.** To induce a low level of replication stress similar to that encountered endogenously, we treated mid-S phase U2OS cells with 100 nM of camptothecin (CPT), a drug known to generate seDSBs by trapping Topoisomerase I (TopI) ahead of RFs. This results in collision of the fork with the trapped cleavage complex, converting the TopI-generated single-strand break into a seDSB[4,5] (Fig. 1a). Pulse labeling of DNA with ethynyl deoxyuridine (EdU) allows visualization of nascent DNA (naDNA) using a post-fixation copper catalyzed "click" reaction to directly conjugate fluorophores to incorporated EdU[33]. By adding EdU to the culture medium contemporaneously with CPT damage, we established a marker for all active RFs, a subset of which would colocalize with seDSBs and their subsequent repair. In combination with EdU pulse labeling for detection of naDNA, we also used bromodeoxyuridine (BrdU) incorporation and immuno-detection without DNA denaturation to detect ssDNA[34]. To visualize repair-associated proteins within the dense environment of the nucleus, we optimized a pre-extraction step prior to fixation that resulted in the removal of the majority of the cytoplasm and soluble nuclear fraction while maintaining intact chromatin structure together with its bound proteins[35,36]. This allowed straightforward immunolabeling of repair proteins and histone modifications to be conducted together with naDNA and ssDNA detection. To assess the generation of seDSBs, we analyzed control and CPT-treated cells using a comet assay (Supplementary Fig. 1a) and confirmed DSB induction. We ensured that the repair foci we monitor are indeed seDSB and not stalled RFs[37–40] by treating S-phase cells with a mild dose of hydroxyurea (HU)—causing RF stalling but not seDSB[41]—and monitored the time course for association of key proteins thought to be involved in both stalled RF rescue and HR (RPA, BRCA2, RAD51, and RAD52). These showed that, after 90 min recovery from the HU treatment, RF colocalization of these proteins had returned to control levels (Supplementary Fig. 1b). We therefore focused our experiments on cells that were allowed to recover for at

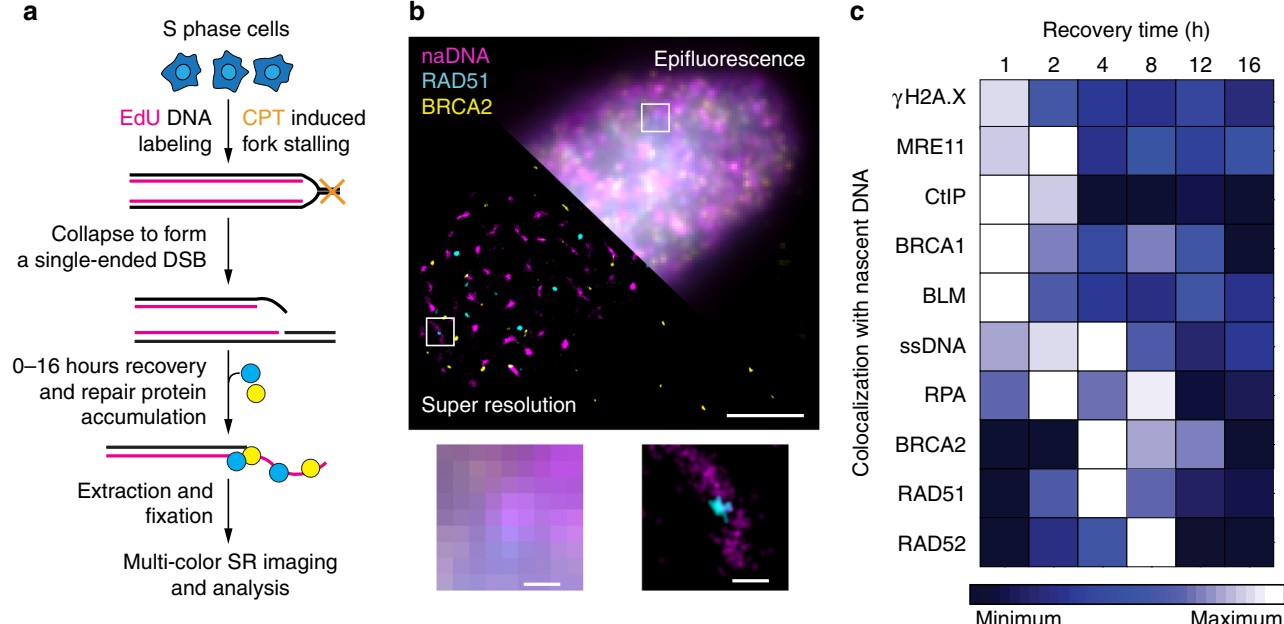

**Fig. 1** Super-resolution spatiotemporal mapping of the arrivals, accumulations, and departures of repair proteins at individual seDSBs in human cells. **a** Damage and labeling scheme used to generate and visualize seDSB repair foci. **b** Representative epifluorescence (upper right) and super-resolution (lower left) images of a single nucleus damaged and labeled for naDNA (using EdU, magenta), RAD51 (cyan), and BRCA2 (yellow). Representative zoomed in epifluorescence (bottom left) and super-resolution (bottom right) foci shown. Whole-cell image scale bar = 3 μm, zoomed sections = 250 nm. **c** Spatiotemporal heatmap describing the arrival, accumulations, and departure kinetics of repair proteins, as well as ssDNA and histone modification γH2A. X, over 16 h of recovery. For complete *N* values, see Supplementary Table 1. For Student's *t* test analysis for significance, see Supplementary Fig. 6

least 1 h before imaging, as repair foci in these cells will only constitute seDSBs, since any RF stalling events would already be resolved.

To visualize individual repair foci at the single-molecule level, we used multicolor stochastic optical reconstruction microscopy[42,43] (Fig. 1b). By removing both CPT and EdU from the culture medium and allowing the cells to recover for between 0 and 16 h prior to fixation and imaging, we generated snapshots of the HR process. The degree of colocalization between naDNA, ssDNA, and proteins was calculated for each of these images by generating an averaged Monte Carlo randomized simulation of each individual cell for normalization (Supplementary Fig. 2). Colocalization in damaged cells was then further assessed by comparison with colocalization levels in control cells, with RPA or RAD51 foci formation used to monitor resection progression[44–47]. To demonstrate the efficacy of area of RPA overlap as a good measure of resection, we treated cells with Mirin, an MRE11 nuclease inhibitor, and detected almost complete abrogation of RPA association with the damage foci (Supplementary Fig. 1c).

By assessing colocalization of various HR proteins with naDNA foci over 16 h of recovery from DSB induction, we were thus able to map the arrivals, accumulations, and departures of repair proteins throughout HR (Fig. 1c). This analysis yielded immediate insights into HRR in vivo at low damage levels. Surprisingly, despite repair taking 8–12 h based on the persistence of RAD51 and BRCA2—two proteins known to be involved in late HR—the amount of colocalized γH2A.X diminished after only 2 h, indicating its removal or relocation away from the DSB DNA prior to a successful repair outcome. The resection nuclease MRE11, its mediators CtIP and BRCA1, and a known HR helicase, BLM, were all detected most strongly 1–2 h following damage and persisted at lower levels—although still slightly higher than in control cells—throughout the 16 h of repair. Both ssDNA and RPA detection were shown to be good indicators of resection, with peak areas of colocalization 2–4 h after damage. On the other hand, RAD51,

RAD52, and BRCA2 localized to the repair foci at 4–8 h. This progression is in general agreement with the theoretical temporal map of HR. Therefore, to further interrogate our data, we developed computational approaches to define parameters describing protein–protein associations at naDNA foci. These parameters included dependence, exclusion, and prevalence, as well as spatially proximal or distal associations of the various repair factors at individual forks (Supplementary Fig. 2).

**RAD51/RAD52 colocalize and displace RPA prior to BRCA2 arrival.** After resection commenced, as detected by increased RPA and ssDNA signal (0–2 h following damage), RAD51 association with repair foci increased drastically between 1 and 2 h of recovery, plateauing at 8 h and then diminishing by 12–16 h. This trend agrees well with the expected sequential ordering of resection occurring prior to RAD51 nucleofilament formation. Similarly, RAD52 colocalization was elevated by 2–4 h and peaked after 8 h before diminishing alongside RAD51 at 12–16 h at which time we hypothesize that the majority of repair had been completed (Fig. 2a, b). Importantly, BRCA2 was not detected until 4 h into recovery by which time a substantial amount of RAD51 had already accumulated at damage foci, indicating some degree of ssDNA/RAD51 nucleofilament formation before BRCA2 association (Fig. 2a–c). These unique RAD51/RAD52 and RAD51/BRCA2 localization trends at 2 and 4 h of recovery are readily observed in the SR images of cell nuclei and individual seDSB foci as shown in Fig. 2d–h; At 2 h, >60% of the repair foci contain both RAD51 and RAD52 (Fig. 2d, e), while co-analysis of BRCA2 and RAD51 shows a lack of BRCA2 at the foci despite the presence of RAD51 (Fig. 2e). By 4 h, BRCA2 could be observed colocalized with RAD51 at repair foci; however, this remained in contrast to RAD52/RAD51 association, which could be observed at both 2- and 4-h recovery (Fig. 2f–h).

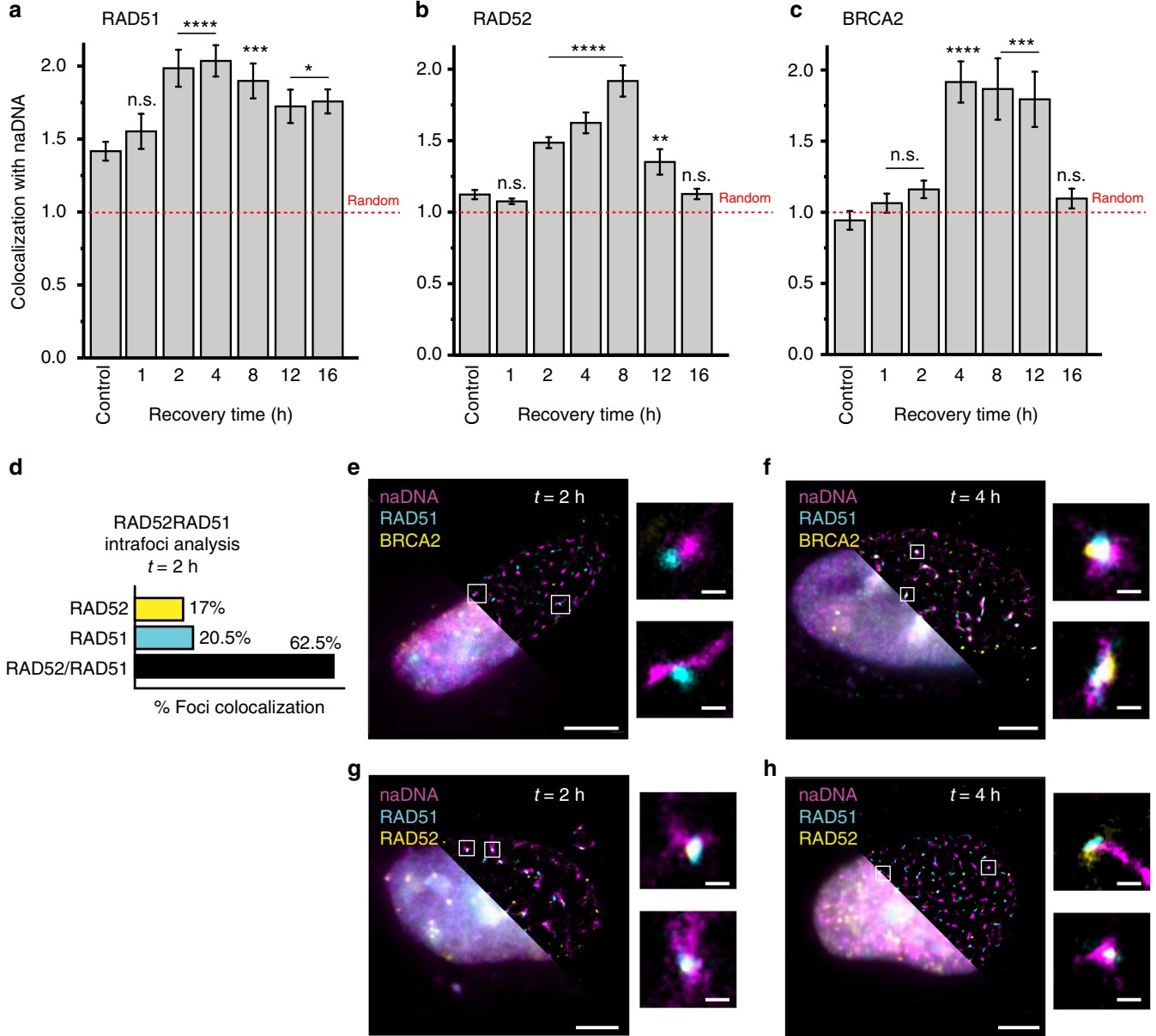

**Fig. 2** RAD52 colocalizes with RAD51 at damage sites prior to BRCA2 colocalization. **a–c** Kinetics of **a** RAD51, **b** RAD52, and **c** BRCA2 colocalization throughout repair (over 16 h of recovery). For complete N values, see Supplementary Table 1. Values were calculated using the Monte Carlo randomization method as described. This allowed the detected number/area of colocalization for each cell to be normalized to the predicted number/area of colocalization in a random simulation of the same cellular image. As plotted here: 1 indicates random overlap (shown as red dashed line), whereas 2 indicates double the number/area of overlaps as expected based on the randomized model. Colocalization in undamaged control cells also shown. Error bars represent mean ± s.e.m. Student's t test for significance between control and damage levels. n.s.p > 0.05, *p < 0.05, **p < 0.01, ***p < 0.001, ****p < 0.0001. **d** Analysis of the colocalization of RAD51 and RAD52 at repair foci 2 h after damage. For complete N values, see Supplementary Table 2. **e–h** Representative whole-cell super-resolution (top right) and diffraction limited (bottom left) images of cells stained for **e**, **f** naDNA, RAD51, and BRCA2 or **g**, **h** naDNA, RAD51, and RAD52 at **e**, **g** 2 h of recovery or **f**, **h** 4 h of recovery. Zoomed in images show representative protein colocalization at naDNA foci. Whole-cell image scale bar = 3 μm, zoomed sections = 250 nm

These observations were of particular interest because, while the Rad52 protein in unicellular eukaryotes is critical for Rad51 function, it is believed that in higher eukaryotes BRCA2 is the principal facilitator of ssDNA/RAD51 nucleofilament formation and function[24]. Moreover, while mammalian Rad52 has previously been detected at ionizing radiation-induced HRR foci[28], no HR-specific role has yet been defined for this protein, not least because no disease phenotype is associated with its homozygous knockout[29,48]. In contrast to the proposed roles for RAD52 and BRCA2, we detected RAD52 alongside RAD51 at repair foci prior to BRCA2 colocalization (Fig. 2c). This also ruled out the possibility that we were observing RAD52-dependent but RAD51-independent break-induced replication (BIR) as has been described previously[49,50]. These observations were noteworthy given that RAD51-dependent HRR has repeatedly been shown to rely on direct BRCA2 mediation of nucleofilament formation[2,6,8,20,21,51,52]. Our observations thus support a loading mechanism that does not require RAD51–BRCA2 interaction at repair foci in order to achieve RAD51 nucleofilament seeding. Elucidation of delayed BRCA2 recruitment contrasts other studies

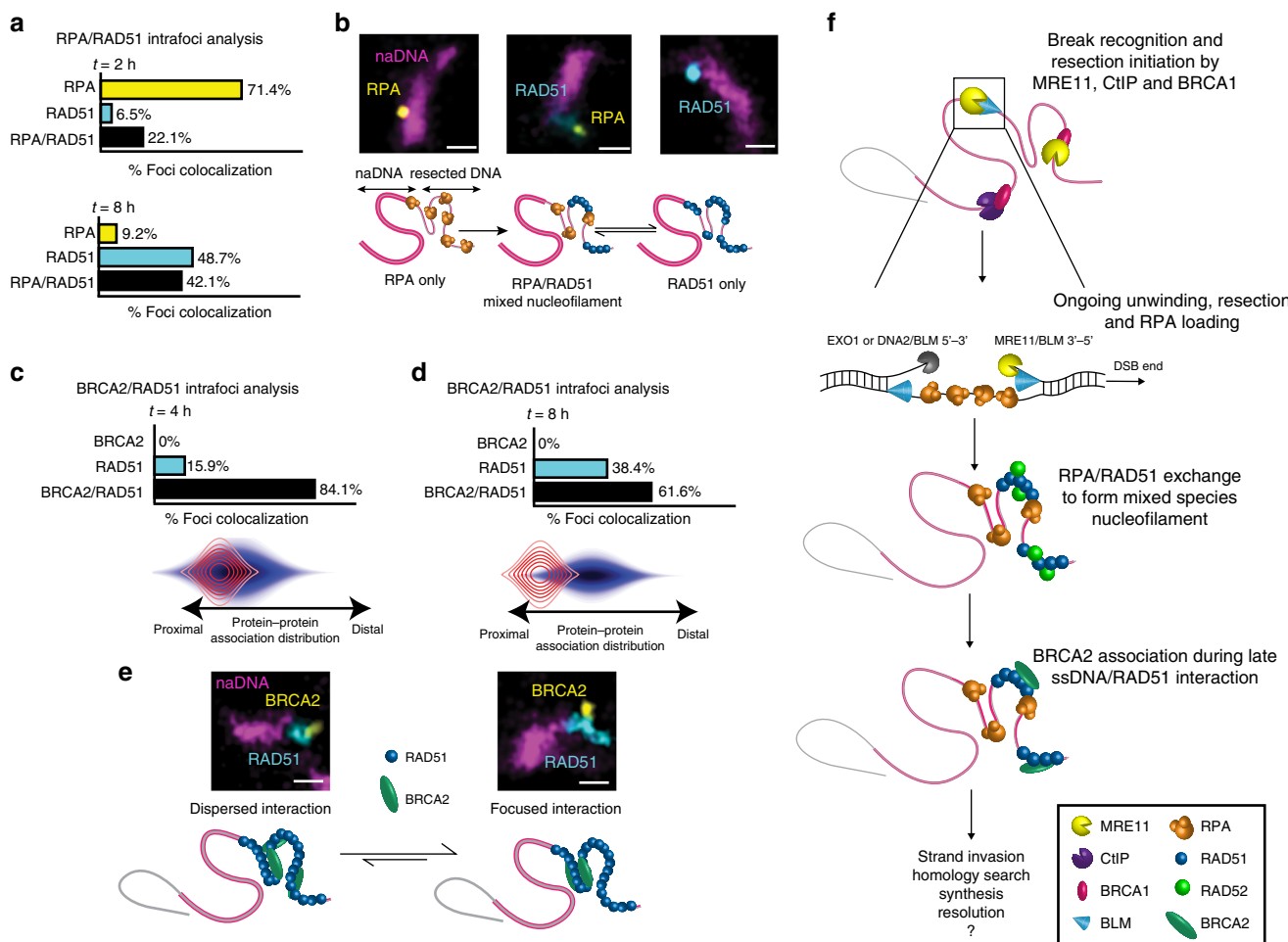

**Fig. 3** RPA dynamically exchanges with RAD51 within the nucleofilament while BRCA2 localizes to RAD51 subpopulations. **a** Analysis of the colocalization of RPA and RAD51 at repair foci 2 and 8 h after damage. **b** Model depicting the observed initial predominance of RPA-only foci before a later transition to equilibrium between mixed RPA/RAD51 and RAD51-only foci. Representative foci shown (scale bar = 250 nm). **c**, **d** Analysis of the colocalization and internal distribution of BRCA2 and RAD51 at repair foci 4 and 8 h after damage. **e** Model depicting earlier dispersed interaction between BRCA2 and the ssDNA/RAD51 nucleofilament and later focused interactions. Representative foci shown (scale bar = 250 nm). **f** A model showing the spatiotemporal associations and interactions of key proteins involved in HR repair of seDSBs, specifically the progression from resection to RPA exchange with RAD51/RAD52, and finally BRCA2 association. For complete N values, see Supplementary Tables 1, 2

that have detected its recruitment to DSBs within 30 s[53] or the first half hour following damage[54]; however, previous observations of sub-minute protein recruitment have generally not only relied on massive, heterogeneous damage induced using laser microirradiation in order to achieve high temporal resolution but also confounding the damage response pathways in play and limiting colocalization to confocal measurements[53,55–57]. Similarly, two-ended DSBs induced by ionizing radiation are not analogous to the HR-repaired seDSBs such as those we examine here and those that are predominantly caused by endogenous stress[4], potentially resulting in different repair pathways. The variance in data likely caused by the confounding variables introduced by laser and ionizing microirradiation were recently demonstrated in a metadata analysis of protein recruitment to damage sites, which showed significant differences between studies[58].

The detection of ssDNA/RAD51 interaction in the absence of direct BRCA2 chaperoning gives particularly significant insight into the HR pathway in vivo. To further assess the crosstalk and interaction of proteins during nucleofilament formation, we analyzed RPA and RAD51 colocalization at repair foci during early nucleofilament formation (2 h) and later during homology search (8 h) (Fig. 3a). Initially, the majority of foci were negative

for RAD51 and showed RPA-only filaments (71.4%), with a subpopulation of mixed RPA/RAD51 filaments (22.1%) and a small fraction showing RAD51-only filaments. By 8 h this population distribution had inverted such that RAD51-only filaments were the dominant species (48.7%), although mixed RPA/RAD51 filaments also persisted (42.1%). During the same time interval (2–8 h), both ssDNA and RPA association plateaued, remaining at high levels before diminishing at 12–16 h (Fig. 1c). A small but significant decrease in RPA signal at 4 h coincided with peak RAD51 colocalization and likely indicates that at this time ssDNA/RAD51 interactions are more prevalent than ssDNA/RPA interactions; however, it is also possible that the decrease in RPA signal is due to changes in the compaction or organization of the resected ssDNA/RPA resulting in a smaller overlap area. These data further confirmed the persistence of RPA in mixed RPA/RAD51 nucleofilaments late into recovery, contradicting the notion that RAD51 fully replaces RPA prior to homology search. Rather than RPA being replaced by RAD51 as is often assumed in the HR pathway[59,60], we conclude that the interactions of these proteins with resected ssDNA are a consequence of an equilibrium in which RPA and RAD51 are often simultaneously present on the resected DNA. This would allow continuous loading and unloading of the two ssDNA

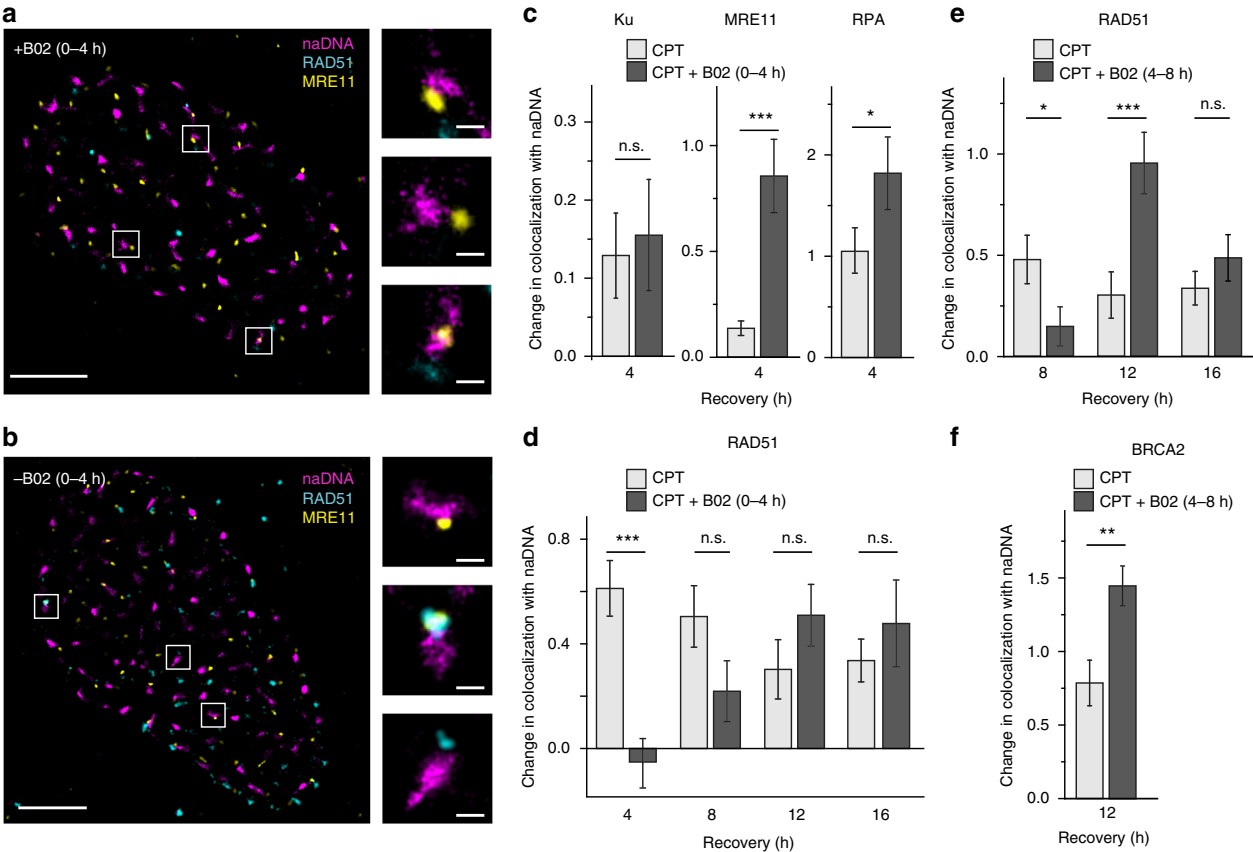

**Fig. 4** The RAD51 inhibitor B02 prolongs resection and antagonizes ssDNA/RAD51 nucleofilament formation and function. **a**, **b** Representative SR images of two nuclei immunolabeled for naDNA (magenta), RAD51 (cyan), and MRE11 (yellow) 4 h after release from CPT damage. The cell in **a** was further treated with B02 during the 4 h of recovery from CPT compared to the control cell shown in **b**. Representative multicolor foci shown. Whole-cell image scale bars = 3 μm, zoomed sections = 250 nm. **c** Quantification of Ku, MRE11, and RPA association with repair foci in cells treated with B02 to inhibit nucleofilament formation during the first 4 h following 1-h CPT treatment. **d** Quantification of RAD51 association with repair foci in cells treated with B02 to inhibit nucleofilament formation during the first 1 h CPT treatment. **e** Quantification of RAD51 association with repair foci in cells treated with B02 to inhibit nucleofilament function beginning 4 h into recovery from CPT treatment for a further 4 h. **f** Quantification of BRCA2 association with repair foci in cells treated with B02 to inhibit nucleofilament function beginning 4 h into recovery from CPT treatment for a further 4 h. For complete $N$ values, see Supplementary Table 1. Error bars represent mean ± s.e.m. Student's $t$ test for significance between CPT and CPT+B02. n.s.$p > 0.05$, *$p < 0.05$, **$p < 0.01$, ***$p < 0.001$

binding proteins, enabling different areas of the resected DNA to be involved in homology search (Fig. 3b)[61,62].

Further examination of the association of BRCA2 with repair foci revealed a dependence on contemporaneous RAD51 association and a prevalence of BRCA2 and RAD51 colocalization over either species individually at both 4 and 8 h (Fig. 3c, d). We also detected a distal spatial relationship between RAD51 and BRCA2 at 4 h that increased in separation by 8 h. These arrangements show that BRCA2 is preferentially associated with subsections of the ssDNA/RAD51 nucleofilament, likely separated by regions of either mixed RPA/RAD51 or of RAD51 devoid of BRCA2 (Fig. 3e). Together, the spatiotemporal mapping of repair protein association with seDSBs yields a dynamic picture of HR in vivo in which extensive and prolonged resection by MRE11, BRCA1, CtIP, and BLM occurs prior to RAD51 association. The formed ssDNA/RAD51 nucleofilament first colocalizes with RAD52 and later with BRCA2, existing as a mixed RPA/RAD51 species for much of the repair process (Fig. 3f).

The above observations are not what would be expected of RAD52 and BRCA2 at HRR foci because of the lack of established role for RAD52 and the long-held belief that RAD51 association depends on BRCA2 mediation. Nevertheless, we successfully confirmed our observations as representative of HR by inhibiting

ssDNA/RAD51 interaction and nucleofilament function using the RAD51 inhibitor B02[63,64]. Treatment of cells with B02 immediately following CPT treatment to inhibit early RAD51 functionality and ssDNA/RAD51 nucleofilament formation increased MRE11 presence at repair foci after 4 h of recovery while decreasing RAD51 colocalization (Fig. 4a, b). Quantification of these colocalizations showed that, while B02 did not lead to persistent Ku association, it did cause a significant increase in MRE11 occupation time and a consequent increase in the amount of resected DNA generated and coated with RPA (Fig. 4c). Successful inhibition of ssDNA/RAD51 interaction was confirmed by observation of minimal RAD51 association with repair foci immediately following B02 removal (at 4 h, Fig. 4d). After 4–12 h of recovery from B02 treatment, colocalization of RAD51 increased to levels similar to those detected in cells treated with CPT alone (at 8–16 h, Fig. 4d). B02 was also used later during recovery from CPT (at 4–8 and 8–12 h) to inhibit homology search and strand invasion. This caused a significant delay in RAD51 and BRCA2 colocalization with repair foci, but we found larger accumulations of both proteins after only 4-h recovery from treatment with B02 (at 12 h, Fig. 4e, f, Supplementary Fig. 3). This demonstrated that RAD51 could neither dissociate nor conduct homology search until after the B02 was removed. In all cases, B02 treatment did not lead to an increase in DDR

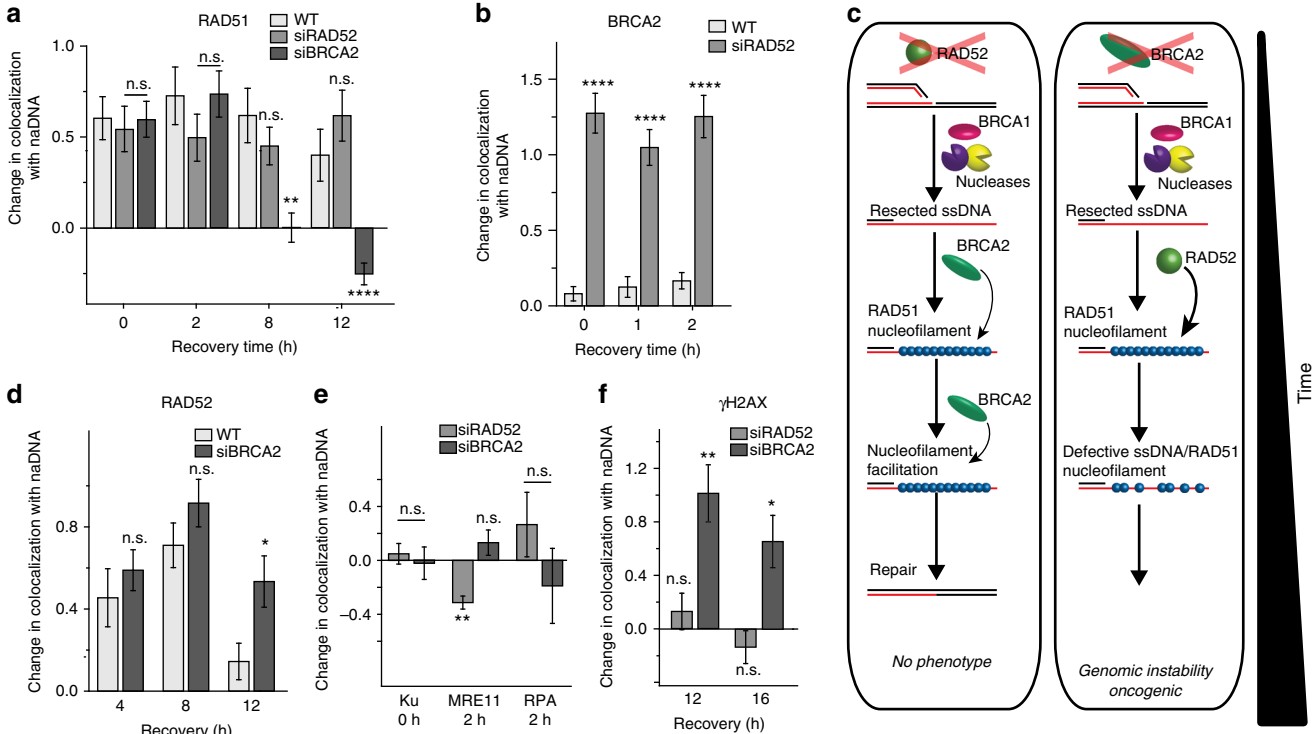

**Fig. 5** Early critical ssDNA/RAD51 interactions can be facilitated by either RAD52 or BRCA2. **a** Quantification of RAD51 association with repair foci during early (0, 2 h) and late (8, 12 h) HR in RAD52- and BRCA2-depleted cells. **b** Quantification of BRCA2 association with repair foci during early (0, 1, 2 h) HR in RAD52-depleted cells. **c** Representations of the HR repair pathway and its deficiencies in RAD52 and BRCA2 depleted cells. **d** Quantification of RAD52 association with repair foci during the homology search stage of HR (4, 8, 12 h) in BRCA2-depleted cells. **e** Quantification of Ku, MRE11, and RPA association with repair foci during early HR (0 or 2 h) in siRAD52- and BRCA2-depleted cells. **f** Quantification of γH2AX association with repair foci during late HR (12, 16 h) in RAD52- and BRCA2-depleted cells. For complete N values, see Supplementary Table 1. Error bars represent mean ± s.e.m. Student's $t$ test for significance between WT+CPT and siRNA+CPT as indicated. $^{n.s.}p > 0.05$, $^*p < 0.05$, $^{**}p < 0.01$, $^{****}p < 0.0001$

signaling as detected by γH2A.X association (Supplementary Fig. 3). Taken together, these observations confirm the spatial and temporal associations of RAD51 and BRCA2 detected in cells treated only with CPT.

**RAD52 facilitates initial ssDNA/RAD51 interactions**. Based on our surprising detection of RAD52/RAD51 association and delayed BRCA2 colocalization, we reasoned that RAD52 might play a role in regulating or mediating early ssDNA/RAD51 interaction independent of BRCA2. We further hypothesized that BRCA2 could potentially function later in HR as a facilitator of nucleofilament function. To test these notions, we examined the effect of RAD52 and BRCA2 knockdown on RAD51 colocalization kinetics. Depletion of BRCA2 did not affect the immediate recruitment of RAD51 to repair foci as detected at 0 h as a significant level of RAD51/naDNA colocalization (Fig. 5a)[40]. This was in agreement with previous studies that describe BRCA2-independent RAD51 association with stalled RFs[23,65]. However, recruitment of RAD51 to seDSBs 2 h following CPT treatment was also detected at similar levels in wild-type (WT) and BRCA2-depleted cells (Fig. 5a), demonstrating that the initial RAD51 recruitment to both stalled and broken forks is BRCA2-independent. Surprisingly, knockdown of RAD52 also had no effect on the early presence of RAD51 (Fig. 5a). However, in siRAD52-treated cells we detected a large increase, relative to WT cells, in BRCA2 association at 0 and 2 h (Fig. 5b).

This increased BRCA2 association in siRAD52-treated cells did not affect the level of RAD51 detected that was elevated following damage to similar degrees in both RAD52-depleted and WT cells (Fig. 5a, b). This result elucidates the ability of BRCA2 to

substitute for the early roles of RAD52 without apparent impairment of the HR pathway. In contrast, RAD52 cannot replace BRCA2 function in BRCA2-depleted cells (Fig. 5c). We conclude this because RAD51 was not detected during later HR (8–12 h) despite a small increase in RAD52 association (Fig. 5a, d). The redundancy between RAD52 and BRCA2 in HRR of seDSBs elegantly explains the synthetic lethality of combined RAD52/BRCA2 deficiencies[26]: rather than RAD52 and BRCA2 functioning in two independent repair pathways, they act sequentially in HR in a manner in which the first and more critical step can be undertaken by either protein. This inference also explains the absence of a strong phenotype in RAD52-deficient mice[48]; the lack of increased damage found in RAD52-depleted cells as demonstrated by the absence of a difference in the amount of Ku, MRE11, and RPA recruited to seDSBs; and the lack of increased damage persistence in RAD52-depleted cells as detected by γH2AX association after 12–16 h (Fig. 5e, f).

**BRCA1 mediates resection and facilitates BRCA2 function.** Combined RAD52/BRCA1 deficiencies have been shown to lead to synthetic lethality similar to that seen in combined RAD52/BRCA2-deficient cells[25], demonstrating the likely existence of crosstalk between BRCA1 and BRCA2. To date, multiple roles have been proposed for BRCA1 in HRR including initial signaling and resection mediation, as well as a potential role upstream of BRCA2 functionality via PALB2 (partner and localizer of BRCA2) binding[16,18,21,66]. However, the in vivo functions of BRCA1 and the relative importance in repair of these mechanisms have been a source of contention for several decades[14,67,68]. Our detection of BRCA1 at repair foci throughout the HR process

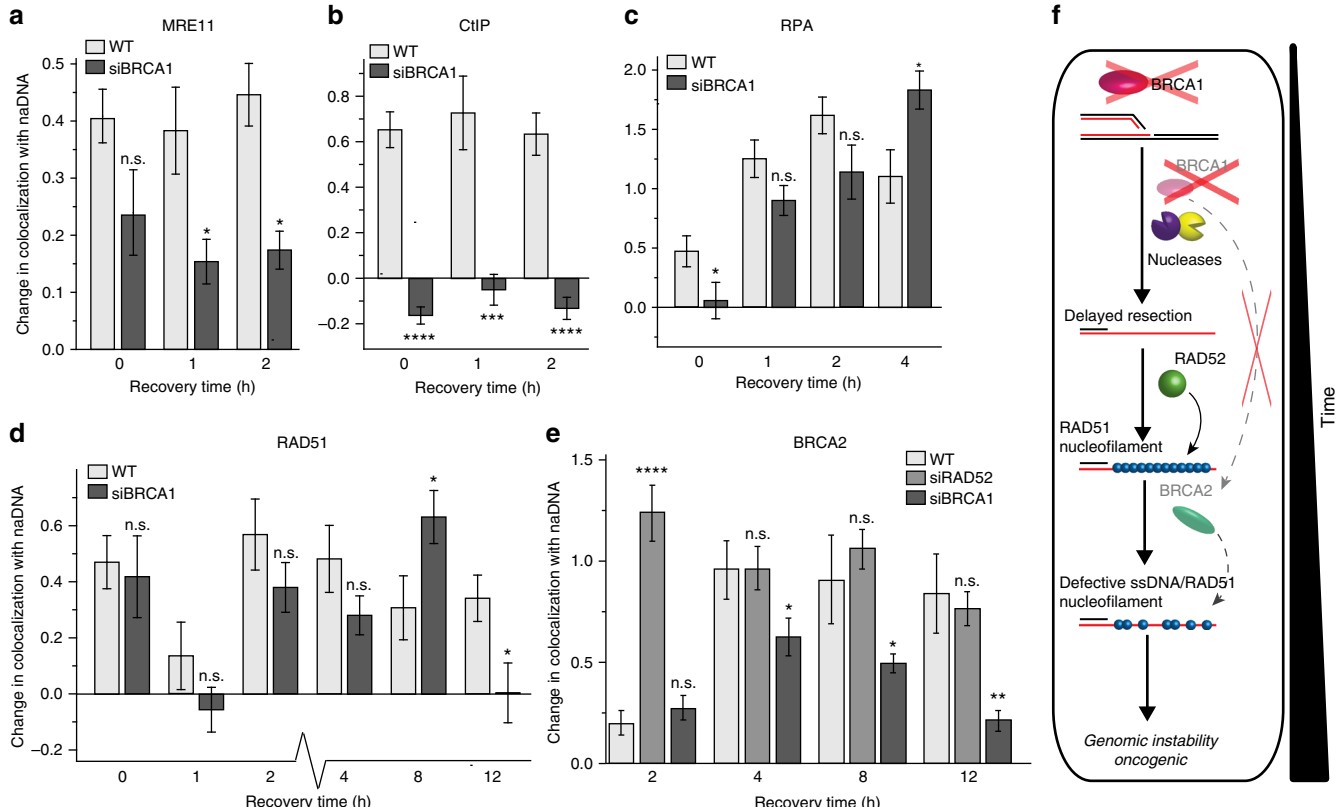

**Fig. 6** BRCA1 is necessary for BRCA2 mediation of ssDNA/RAD51 interactions. **a** Quantification of MRE11 association with repair foci during early HR (0, 1, 2 h) in BRCA1-depleted cells. **b** Quantification of CtIP association with repair foci during early HR (0, 1, 2 h) in BRCA1-depleted cells. **c** Quantification of RPA association with repair foci during early HR (0, 1, 2, 4 h) in BRCA1-depleted cells. **d** Quantification of RAD51 association with repair foci throughout HR (0, 1, 2, 4, 8, 12 h) in BRCA1-depleted cells. **e** Quantification of BRCA2 association with repair foci during HR (2, 4, 8, 12 h) in BRCA1- and RAD52-depleted cells. **f** A representation of the HR repair pathway deficiency in BRCA1 depleted cells. For complete *N* values, see Supplementary Table 1. Error bars represent mean ± s.e.m. Student's *t* test for significance between WT+CPT and either siBRCA1+CPT or siRAD52+CPT as indicated. [n.s.]*p* > 0.05, \**p* < 0.05, \*\**p* < 0.01, \*\*\**p* < 0.001, \*\*\*\**p* < 0.0001

(Fig. 1c) demonstrates the likelihood of multiple roles, including beyond initial DDR signaling and resection, and possibly related to RAD51/BRCA2 interaction and homology search. To resolve the multiple roles of BRCA1 in vivo and to assess their importance, we imaged the localization kinetics of repair proteins in BRCA1-depleted cells. We found that BRCA1 knockdown produced inhibition of early HR by diminishing MRE11 association and blocking CtIP recruitment (Fig. 6a, b). This led to a significant delay in resection, with less RPA recruitment detected in siBRCA1 cells compared to in WT cells immediately following damage (Fig. 6c). However, RPA association with repair foci reached WT damaged cell levels after a 2-h delay (i.e., after 4 h of recovery from CPT) (Fig. 6c)[69]. Surprisingly, RAD51 levels were also comparable to those in WT cells, albeit also delayed, reaching a peak at 8 h instead of at 2–4 h (Fig. 6d). We believe that this result demonstrates that resection was at least partially successful in BRCA1-depleted cells, allowing initial ssDNA/RAD51 nucleofilament formation to proceed by 8 h. While it is possible that the loss of other protective and reparative BRCA1-dependent pathways induced higher levels of damage (as detected via comet assay, Supplementary Fig. 4), delayed recruitment of RAD51 in the absence of BRCA1 does not seem to indicate this. Furthermore, BRCA2 association—while not fully diminished at 8 h of recovery—did not reach WT levels at any time point, and RAD51 association also decreased more quickly than expected based on the WT spatiotemporal map (Fig. 6d, e). Moreover, despite successful resection and RAD51 loading, BRCA1-depleted cells displayed persistent damage similar to that found in BRCA2-depleted cells (Supplementary Fig. 4). These observations are in agreement with the documented genomic instability characteristic of BRCA1- and BRCA2-deficient cells and of mutant disease phenotypes[19]. We infer that the enhanced DNA damage detected in BRCA1-depleted cells is primarily a consequence of downstream defective BRCA2-mediated ssDNA/RAD51 interaction (Fig. 6f).

To further test this conclusion, we examined double knockdown siBRCA1/siRAD52-treated cells to determine whether BRCA1 deficiency would affect the BRCA2-dependent ssDNA/RAD51 interactions necessary for initial nucleofilament formation in the absence of RAD52. Indeed, SR imaging of damaged BRCA1/RAD52-depleted cells revealed no detectable association of either BRCA2 or RAD51 with stalled RFs or seDSB repair foci (Fig. 7a, b). These double-depleted cells also displayed the characteristic synthetic lethality and DSB sensitivity previously reported[25]; 4 h after CPT treatment, the cells were no longer suitable for SR imaging because of diminished cell viability. Attempts to prepare double-depleted siRAD52/siBRCA2 cells similarly demonstrated this synthetic lethality with most cells succumbing to apoptosis during the transfection process. We conclude that, in the absence of sufficient RAD52 to mediate RAD51 function together with inhibition of BRCA2, there is a lack of RAD51-dependent protection of RFs and a resultant production of more seDSBs[23,65,70]. The lack of BRCA2 facilitation of ssDNA/RAD51 interaction must be a consequence of BRCA1 depletion: if BRCA1 played a direct role in RF protection, the same effect would have been observed in BRCA1-depleted

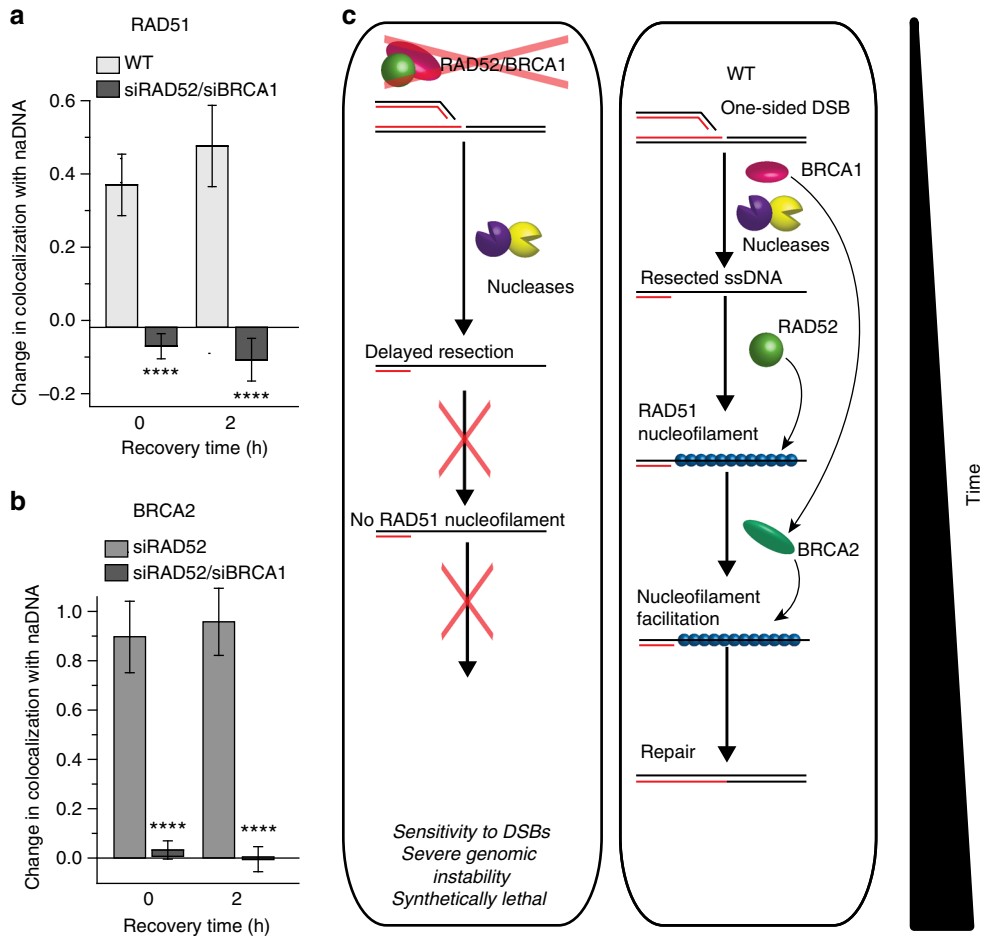

**Fig. 7** In the absence of both RAD52 and BRCA2, ssDNA/RAD51 interactions are severely abrogated. **a** Quantification of RAD51 association with repair foci following CPT treatment and release (0, 2 h) in BRCA1/RAD52-depleted cells. **b** Quantification of BRCA2 association with repair foci following CPT treatment and release (0, 2 h) in RAD52- and BRCA1/RAD52-depleted cells (RAD52-depleted cells shown to contrast increased BRCA2 levels required for ssDNA/RAD51 interactions). **c** Representations of the HR repair pathway deficiency in RAD52/BRCA1-depleted cells and the concluded HR repair pathway in vivo in WT cells. For complete *N* values, see Supplementary Table 1. Error bars represent mean ± s.e.m. Student's *t* test for significance between WT + CPT and siRAD52/siBRCA1+CPT and siRAD52+CPT and siRAD52/siBRCA1+CPT as indicated. ****$p < 0.0001$

cells. This collapse of RFs into seDSBs should not have proven lethal in BRCA1/RAD52-depleted cells. However, despite sufficient resection ability (as detected in BRCA1-depleted cells (Fig. 6)), we found no evidence of the initial ssDNA/RAD51 interactions required for later nucleofilament formation and HRR. This is because BRCA1 depletion abrogates BRCA2 functionality, and this, combined with RAD52 depletion, results in the absence of a critically needed RAD51 mediator (Fig. 7c).

## Discussion

Using SR imaging, we have visualized the collapse of RFs to form seDSBs and their subsequent repair over 16 h of recovery. Because of the inherent single-molecule sensitivity of these assays, we were able to examine these breaks at a relatively low level of induction[71,72] and to discern small changes in protein association and intra-foci structure. This allowed us to define the progression of HR in unprecedented detail at break sites similar to those generated endogenously, revealing several previously uncharacterized interactions, dependencies, and inter-relationships within this critical pathway. These findings will form the basis for future work incorporating other molecular techniques to further reveal the in vivo nature of BRCA1, BRCA2, RAD51, and RAD52 and their crosstalk.

Our analysis of RAD51 recruitment, its presence at naDNA, and its correlation with other key proteins raises several key issues with respect to the formation of recombinase complexes in HR. Current models describe a linear process consisting of the following discrete consecutive steps: resection, RPA loading onto ssDNA, removal of RPA, and BRCA2-dependent loading of RAD51. Here we show that these steps coexist, in particular through the reversible interchange of RAD51 with RPA and its persistent interaction with RAD52 (Figs. 2 and 3). Surprisingly, our mapping of HR also demonstrates that RAD51 nucleofilament formation commences without BRCA2 colocalization at the DSB; BRCA2 arrives later and operates on subsections of the nucleofilament throughout homology search and eventual repair resolution (Fig. 2a, c). Based on these insights, we were able to define the dynamic associations and relationships of key HR proteins at collapsed RF-associated seDSBs and establish a refined model for the spatiotemporal kinetics of their repair (Fig. 3f). Furthermore, by investigating the crosstalk between RAD51, RAD52, BRCA1, and BRCA2, we have also defined their individual roles and interactions within the pathway (Fig. 7c). In particular, by comparing the spatiotemporal associations of proteins in cells depleted of BRCA2 or RAD52, we uncovered the hitherto unknown function of RAD52 in enabling initial ssDNA/RAD51

interactions. The fact that BRCA2 can fully replace RAD52 in this early role explains the lack of a strong disease phenotype in knockout mouse models[48] and highlights the previous difficulty in identifying this important step in the HR process. We also demonstrate that the later requirement for BRCA2 facilitation of ssDNA/RAD51 function cannot be carried out by RAD52, thus explaining the genomic instability inherent in BRCA2-deficient cells (Fig. 5c). Although BRCA1 depletion affects both resection and BRCA2 function (Fig. 6), upstream BRCA1 activity is critical for BRCA2 mediation of later ssDNA/RAD51 function, as well as early ssDNA/RAD51 interaction in RAD52-depleted cells. Surprisingly, although we detect BRCA1 at DSBs during resection and there is ample literature demonstrating its resection mediation abilities[8,18,21], by examining BRCA1-depleted cells at stress levels lower than most previous DDR studies, we detect a delay, but no comprehensive deficiency, in resection because both RPA and RAD51 coating of the resected ssDNA reach levels similar to those detected in WT damaged cells. This is further explained by recent studies that identified CtIP–BRCA1 interaction as non-essential for resection but capable of speeding it up[73,74]. At higher damage levels, it is possible that resection cannot similarly be delayed without further consequences due to limitations on the total amount of repair proteins present and the threshold for damage amount and duration.

This highlights our motivation for combining innovative damage and labeling assays with SR imaging: it allowed for a comparatively low level of DSB induction to be examined with minimal interference from alternate damage motifs or repair pathways. For this reason, we used CPT instead of ionizing radiation, which is a more common damage induction method but one that causes clustered, heterogeneous damage[58], quite unlike the sparse endogenous seDSBs that are specifically repaired via canonical HR and well mimicked by CPT DSB induction[4]. Importantly, and in light of previous studies that have shown CPT causes both RF stalling and DSB induction[37], we used comet assays to confirm DSB induction and compared CPT-induced damage with HU-induced damage and consequently chose to focus on repair foci that persisted for >60 min in order to minimize signal from stalled, but unbroken RFs (Supplementary Figs. 1 and 4).

Despite evidence that HR is the predominant repair pathways for seDSBs in S-phase cells, we also considered the possibility that we were observing RAD52-dependent/RAD51-independent BIR as has been described[49,50]. Discerning BIR from canonical HR remains challenging; however, the persistence of RAD51 at RAD52-positive repair foci made it unlikely that we were detecting a RAD51-independent pathway.

Our surprising results regarding RAD52's role in HR as a RAD51 mediator also prompted us to consider previous studies that identified RAD52 as an antagonist of stalled RF stability[70], whereas BRCA2 is a known mediator of RF protection[75]. We thus considered the possibility that the BRCA2 association we detected in siRAD52-treated cells (Fig. 5b) was due to its recruitment to unbroken RFs in the absence of RAD52-mediated degradation. However, this did not explain BRCA2's colocalization with DSB-specific proteins such as RPA, RAD51, and BRCA1, nor did it account for BRCA2's persistence over >8 h of recovery. Furthermore, BRCA2-dependent protection of RFs should have resulted in a decrease in the number of breaks generated, which we could not detect via comet assay (Supplementary Fig. 4).

A further discrepancy between our data and previous studies is the lack of clustering of repair foci[76,77]. While a large degree of heterogeneity in both individual foci structure and overall foci distribution within a cell was observed during our experiments (for examples, see Figs. 1b, 2e–h, and 4a, b), no obvious overall reorganization of DNA repair foci was witnessed. Quantification of the total number of naDNA/RAD51 foci per cell also failed to

reveal the expected sharp reduction in foci count predicted by the repair factory hypothesis (Supplementary Fig. 7). We suggest that this is again due to the low-level, pseudo-endogenous damage that we have examined. Large chromatin rearrangements have previously been seen in studies causing clustered damage or at least generating two-ended DSBs. Here we have only induced a handful of seDSBs and, in doing so, have not triggered repair pathways that make use of repair foci clustering.

The tenfold enhancement in lateral directions afforded by SR imaging was critical in enabling our novel insights into the HR pathway because it allowed us to visualize spatially separated single DSB repair foci and, in doing so, differentiate recruitment of proteins near the foci from closer associations and interactions. These distinctions cannot be made in diffraction-limited images; however, here we are able to detect the delayed associations of RPA, RAD51, and BRCA2. These observations match the biochemically determined HR timeline[3] but are not in agreement with the less well-defined recruitment kinetics of HR proteins to clustered damage sites that have been imaged conventionally[78–80]. A further advantage of the SR single-molecule imaging approach is its ability to detect low levels or transient associations. This is because every fluorophore detected is counted as equal; thus, repair foci can be examined without thresholding away small or sparse foci. For this reason, our unexpected observations of early BRCA2-negative RAD51 associations (Fig. 2a–c) prompted us to hypothesize that the colocalization we detected was of a more intimate and small-scale nature than what has been conventionally detected using immunofluorescence (IF). Indeed, while conventional imaging can detect long, stably formed RAD51/ssDNA nucleofilaments as well as en masse recruitment of RAD51 to clustered damage sites, it is unlikely that dynamic association of smaller amounts of RAD51 at repair foci could be visualized without SR. This highlights the strengths of single-molecule SR imaging over conventional IF approaches; however, we also considered the limitations and difficulties engendered by SR imaging, particularly in interpreting and presenting data.

This is, in part, due to the fact that single-molecule SR imaging does not produce a true optical image but rather a list of molecular localization coordinates that can then be "rendered" into a pseudo-image. To achieve this, pixels are rendered, typically at 5–20 nm, with the calculated localization precision of each molecule (usually 5–30 nm, depending on background/auto-fluorescence, imaging aberrations, and fluorophore brightness) causing its localization to be spread over several of these 5–20-nm pixels. This is a fundamental difference in the nature of images produced from single-molecule SR compared to confocal and epifluorescence microscopies and must be considered when analyzing and presenting SR data.

One the one hand, this is an important advantage of SR because it allows for analysis of subpopulations and a holistic approach to mapping distributions. This specifically contrasts with confocal approaches that typically focus on the brightest areas, particularly when studying DDR and considering DNA damage foci. Such clustered foci in confocal microscopy allow all other non-clustered proteins to be thresholded out of the image, whereas in SR analysis these unclustered proteins are given equal weighting to individual proteins within clustered foci and are not so easily ignored, presenting a potential difficulty in analysis. A similar effect has also been observed in SR imaging of microtubules with a large number of non-microtubule-associated fluorophores previously rendered and assessed as "non-specific." In reality, these localizations are true tubulin signals from soluble tubulin dimers within the cell[81]. Importantly, the ability to detect individual fluorophores and their underlying antigens has allowed us, within this study, to examine individual DSBs at very low

levels of damage, but it has required more complex analytical tools and resulted in images that are quite different to confocal DDR micrographs.

Specifically, many of the immunolabeled DDR proteins we have imaged and shown throughout our paper are not colocalized with naDNA or each other. Rather than this being demonstrative of non-specific binding or "weak" DDR signal, it is because these localized proteins are, individually, not participating within the DDR at the time of imaging, exactly as one would expect.

With hundreds to thousands of individual protein signals (detected as clustered fluorophore localizations attributable to either single proteins or multiple copies) detected in each cell, the rendered pseudo-images did not lend themselves to straightforward overlap quantification. Thus, in order to quantify association between naDNA, proteins, and γH2A.X, we used a Monte Carlo simulation to randomly redistribute clusters of one color within the nuclear region of interest (ROI) while maintaining the "real" distribution of the second color. This allowed us to calculate the number, area, and percentage of overlaps expected from a randomly associated pair of species specific to each cell imaged. This normalized each simulation to both area and to cell-to-cell differences in naDNA and protein density. To generate a colocalization coefficient that described the association relative to random, the number or area of overlaps in the real image was ratioed to an average taken from 20 simulations of the same cell. In this context, a colocalization coefficient of 2 showed a doubling in the number or area of overlaps above random, and 3 showed a tripling. Colocalization coefficients calculated for undamaged control cells were routinely determined to be above random due to HR protein associations with endogenously stressed and broken RFs and also because of their transient binding to unwound DNA. For example, in Fig. 2a, control levels of RAD51 overlap with naDNA were significantly higher than random (colocalization of ~1.5 compared to 1 when random) while RAD52 was slightly higher. Accordingly, we determined changes in colocalization comparative to control levels—for example, after 1 h of recovery, both RAD51 and RAD52 were detected at control levels, before significantly accumulating above control levels in the following hour of repair and remained prevalently colocalized with the naDNA foci until >12 h of recovery (Fig. 2a, b). In only a handful of cases were specific proteins detected as actively excluded from undamaged replication foci (for example, BRCA2, Fig. 2c).

Controls levels are thus crucial in assessing changes in associations, whereas the randomized simulation approach is more important in normalizing for differences in cell size, cluster size, cluster number, etc. Effectively, we are normalizing each cell's overlaps to its own eccentricities, allowing for sensitive detection of changes across the cell population upon DDR. We have previously published an extensive justification and discussion of this method, as well as the associated calculations and ImageJ plugin[82].

While we find it advantageous to be able to visualize the distribution of a protein of interest at a single-molecule level and without the brightness bias of confocal and to examine subpopulations in low damage conditions, the scattered images can also be difficult to present due to the relatively low degree of colocalization. An even more perplexing complication arises due to the enormous number of pixels within a rendered SR image. At 10 nm pixel size, a 25-μm-wide cell will render at 2500 pixels with single protein fluorescent signals routinely confined to 3–4 pixels (~30–40 nm due to antibody size and localization error). However, journals routinely print at 300 dpi and cell images often only occupy one or two inches. This convolves 4–8 pixels of signal into a single printed pixel, which is almost impossible to see with the naked eye. Because of this, SR images are brightened and fluorescent signals are dilated and smoothed to make them visible. However, this is not a real representation of the image, which,

itself, is also only a rendering of a coordinate list. For this reason, and the marked differences with confocal images, presented images must be considered descriptive only.

Careful optimization of the conditions used for extraction and fixation of samples[81] was also key in detecting protein associations predominantly associated with CPT damaged foci. To this end, we used a protocol whereby the majority of the cytoplasm was removed as well as a significant amount of the soluble nuclear fraction, thus leaving a sample consisting largely of insoluble chromatin-associated proteins[83,84]. This minimized the proportion of SR signal originating from repair factors not involved in the HR process and allowed sufficient statistical significance to describe the interactions and the kinetic progressions with better acuity than previously achieved. It is worth noting that our reliance, as well as that of many others, on immunolabeling inevitably contributes to the noise level of SR images. Because SR rendering does not make use of thresholding of sparsely labeled foci but rather treats every detected molecule as equal, non-specific binding of antibodies creates localizations in SR renderings that are difficult to identify as artifactual[85]. Moreover, steric exclusion of antibodies due to the density of the target structure and competition between antibodies also dictates that labeling cannot be considered comprehensive. This is more problematic for SR because of its ability to image at resolutions approximately the size of single antibodies. Both non-specificity and steric exclusions inevitably lower the sensitivity of the SR assay.

We also considered whether differences in antibody sensitivity could potentially introduce bias in our analyses toward detection of the more sensitively detected proteins. In responses to this potential limitation, the results reported here focus primarily on analyses that we determined would not be affected by sensitivity differences; both the quantification of protein overlap with naDNA foci and the protein–protein association distribution avoid this problem. In the former case, the level of overlap of a particular protein is normalized to control levels of overlap and to random simulations, both of which carry the same inherent antibody sensitivity and therefore, with normalization, remove this potential confounding error. In the case of the spatial analysis, both target proteins must be labeled in order for the analysis to be carried out so there is no potential bias here to a more or less sensitive antibody. An error would only arise in this analysis if an entirely insensitive (absent or non-specific) antibody was used, something we avoided by using antibodies that had been validated. We acknowledge that the intrafoci analysis (Figs. 2d and 3a, c, d) is susceptible to bias from differences in antibody sensitivity, and for this reason, we have limited ourselves to qualitative conclusions drawn from these analyses that could be strengthened by other observations. With more specific approaches to labeling, we envision an increase in sensitivity allowing for smaller-scale changes in the accumulation of proteins such as RAD51 and RPA to be monitored, as well as quantitative intrafoci analysis. Furthermore, stoichiometric labeling will allow quantification of HR proteins at these single DSBs[86,87].

The novel and modular assays developed in this study should be of particular interest to those studying genomic stress and stability and will provide new insights into various pathways and the proteins involved in the DDR[35,36,41,88]. This is because they allow detection and observation of single DSB events in the densely populated and noisy nuclear environment, a feat previously hindered by the diffraction limit of light and an inability to confidently identify single DSBs in intact cells. We have overcome these issues by combining SR with low doses of CPT, which causes RFs to collapse into seDSBs[4,89]. To image DSBs, we combined "click" labeling of naDNA[33,90] with dual-color immunolabeling. Because crosstalk between repair pathways and signals from the low level of endogenous damage were inevitable, we strove to work at a DSB-induction level sufficient to

detect the progression of HR at synchronized CPT-induced breaks (a successful endeavor as evidenced by the spatiotemporal kinetic map in Fig. 1c) but significantly lower than previously used for immunofluorescent visualization[71,72]. Importantly, we also show that CPT-induced RF stalling/regression[37] did not interfere with HR observations beyond the first 90 min of repair but that RAD51/RAD52 might also be involved in RF protection (Supplementary Fig. 2, Figs. 5a and 7a, b). While the research outlined here focused on a handful of specific HR proteins, the assays we describe can be extended to any protein of interest that can be fluorescently labeled. As such, we provide an important new tool capable of mapping specific molecular complexes involved in DDR. This will complement the extensive biochemical approaches currently used to identify and characterize these proteins.

Moreover, our observations highlight the critical nature of initial ssDNA/RAD51 interactions, particularly when compared to the less severe consequences of deficiencies in BRCA1's role in resection or BRCA2's mediation of later ssDNA/RAD51 function. While cells that are subject to late HR defects (via either BRCA1 or BRCA2 mutation/depletion) are DSB sensitive, cells unable to mediate initial ssDNA/RAD51 interactions display the far more severe phenotypes of synthetic lethality, undergoing rapid apoptosis[25,26]. Our data explain the similar phenotypes seen in both BRCA1- and BRCA2-deficient cells, as well as the diminished levels of BRCA2 and RAD51 seen during late HR upon BRCA1 depletion. The redundancy between RAD52 and BRCA2 in mammalian cells is in agreement with the previous observation that both proteins are related to Rad52 in budding yeast, a species that lacks a BRCA2 homolog and that relies entirely on Rad52 for successful HR[91]. Our observations define a division of function between RAD52 and BRCA2 that opens up new avenues of enquiry into potential differences in HR pathways between simple organisms like yeast and multicellular eukaryotes containing far larger genomes that might require greater stabilization during later HR. Finally, elucidation of the mechanisms underlying the synthetic lethality between RAD52 and BRCA1/2[24–26] could identify potentially attractive therapeutic approaches for cancers in which BRCA1/2 is mutated.

## Methods

**Cell synchronization and drug treatment**. Female human bone osteosarcoma U2OS cells (ATCC HTB-96) were cultured in McCoy's 5A (Modified) medium (ThermoFisher 16600) with 10% fetal bovine serum (FBS; Gemini Bio. 100-106) and 100 U/mL Penicillin–Streptomycin (ThermoFisher 15140) at 37 °C and 5% $CO_2$. Cells were trypsinized and seeded at low density on glass coverslips in 6-well plates and allowed to adhere in complete medium for 18–24 h before being switched to FBS-free medium for a further 48–72 h. This synchronized cells in predominantly G0/G1 phase[92]. Cells were then released in complete medium for a further 16 h to produce a majority mid-S-phase population. To generate seDSBs, cells were treated with 100 nM CPT (Abcam 120115)[4] for 1 h. Coincident with CPT treatment, cells were also fed 10 μM EdU, a thymidine analog, in order for incorporation into nascent DNA through endogenous replication (Click-iT Kit, ThermoFisher C10340)[41]. Immediately following damage and EdU treatment, cells were fixed in order to access damage and response to DSB generation. To assess recovery time points a further 1, 2, 4, 8, 12, or 16 h after CPT-induced damage, cells were cultured without CPT in full medium. Control cells were treated with 0.1% DMSO in place of CPT. HU experiments were carried out using the same synchronization as for CPT experiments but instead cells were treated with 0.1 mM of HU for 4 h, beginning 15 h after release from starvation. EdU staining was achieved during the final hour of HU treatment.

To further perturb and probe the HRR process of DSBs, U2OS cells were treated with B02 (EMD Millipore, 553525), a RAD51 inhibitor with proposed mechanisms for antagonizing RAD51/ssDNA nucleoprotein filament assembly, disassembly, and strand invasion. B02 was administered to cells following CPT damage to assess changes in HRR involving each of the discrete proposed inhibitory mechanisms[63,64]. First, to probe inhibition of RAD51 binding to resected DNA, B02 was added to the culture medium to a final concentration of 20 μM immediately following CPT damage for the first 4 h of recovery. To assess changes to the HR process, cells were fixed immediately following 4 h of B02 treatment, as well as after up to 12 h recovery in drug-free medium (16 h total recovery time from CPT treatment). To assess B02 inhibition of RAD51/ssDNA nucleoprotein filament strand invasion, cells were treated with CPT and allowed 4 h to recover in complete medium before addition of 20 μM B02 for a further 4 h. Cells were then fixed (8 h after CPT treatment, immediately following 4 h B02 treatment) or allowed further recovery from the B02 treatment (4 or 8 h after B02 treatment and 12 or 16 h after CPT treatment). Finally, to assess late-stage strand invasion and disassembly of RAD51/ssDNA nucleoprotein filaments, cells were allowed normal recovery for 8 h in complete medium after 1-h CPT treatment. B02 was then added for 4 h taking cells to 12 h overall recovery from initial CPT damage. Again, cells were either fixed at this time or given a further 4 h to recover from the B02 treatment.

To inhibit poly ADP-ribose polymerase, cells were cultured in the presence of 100 μM Veliparib (Santa Cruz, 202901) for 24 h prior to recovery from CPT treatment. To do this, cells seeded on coverslips were administered Veliparib during the final 7 h of serum starvation as well as during release in complete medium and in the CPT/EdU medium. Following CPT treatment, cells were recovered for up to 16 h in medium free of both CPT and Veliparib. Similarly, Mirin (Fisher, 319010), an MRE11 inhibitor, was also combined with CPT to probe perturbations to the resection steps of the HR pathway. Cells were treated with 25 μM Mirin for 24 h prior to recovery as described for Veliparib.

**Small interfering RNA (siRNA) and western blotting**. To knockdown BRCA1, BRCA2, and Rad52, 4 μL of Lipofectamine RNAimax (Invitrogen) and 4 μL of the appropriate siRNA, (HS_BRCA1_9, HS_BRCA2_7, and Hs_Rad52_5, all at 10 μM) (Qiagen) were diluted in 100 μL of Opti-MEM medium (Gibco) and incubated at room temperature (RT) for 5 min. One hundred microliters of the siRNA–lipid complex was then added to one well of a 6-well plate containing 50–70% confluent U2OS cells in 1 mL of Penicillin–Streptomycin-free medium. Control cells were incubated similarly with siRNA for luciferase (Dharmacon). This process was repeated 24 h later. After the second siRNA treatment, the cells were harvested for further experiments either after 24 (siRAD52) or 48 (siBRCA1, siBRCA2, and double siRAD52/siBRCA1) hours after this second siRNA treatment. For CPT or Veliparib treatments, siRNA-treated cells were transferred to McCoy's 5A medium and processed as with WT cell experiments.

To test for successful siRNA knockdown, cells were harvested and resuspended using equal volume of 2× Laemmeli sample buffer (1610737, Bio-Rad) containing 5% β-mercaptoethanol (Sigma) and were lysed by heating the samples at 95 °C for 5 min. Protein concentration was measured using a Nano-Drop spectrophotometer (Thermo). Proteins were resolved by sodium dodecyl sulfate (SDS)-polyacrylamide gel electrophoresis on 4–15% TGX gels (Bio-Rad) in 1× Tris-Glycine-SDS buffer and transferred to polyvinylidene difluoride membrane (Millipore). Membranes were blocked with blocking buffer and then incubated with the following primary antibodies: BRCA1 (sc-642, Santa Cruz), BRCA2 (88361 Novus), Rad52 (sc-365341, Santa-Cruz), and proliferating cell nuclear antigen (ab29, Abcam). Membranes were washed 3× with blocking buffer and incubated with secondary antibodies conjugated to horseradish peroxidase or Alexa Flour 750 (Santa Cruz sc-2357, abcam ab7075, invitrogen A-21037 invitrogen A-21039). Blots were detected using an Enhanced Chemiluminescence Detection Kit (GE Healthcare, Bucks, UK) and were developed with a LICOR Odyssey imager. For western blot, comet, and overlap analysis showing successful siRNA knockdowns and minimal perturbation from the siRNA procedure, see Supplementary Fig. 4.

**Extraction and fixation**. In order to minimize signal from unbound protein and potential non-specific antibody labeling, we optimized[81] an extraction and fixation protocol to remove the majority of the cytoplasm and soluble nuclear content[83,84]. Initially, RT CSK buffer (10 mM Hepes, 300 mM Sucrose, 100 mM NaCl, 3 mM $MgCl_2$, and 0.5% Triton X-100, pH = 7.4) was applied for 2–3 min to pre-extract the cells. Importantly, removing the soluble fraction of the nucleus decreased the overall density of proteins for imaging that allowed us to successfully distinguish between individual DSBs and their repair. This pre-extraction also biased those proteins remaining to being chromatin bound and thus more likely involved in repair and relevant to our analyses. Following extraction, cells were fixed in paraformaldehyde (3.7% from 32% electron microscopic (EM) grade, Electron Microscopy Sciences, 15714) and glutaraldehyde (0.3% from 70% EM grade, Sigma-Aldrich, G7776) in phosphate-buffered saline (PBS) at RT for 15 min. Cells were washed three times with PBS and, if required, stored overnight at 4 °C. For fluorescent tagging of the pulse-labeled naDNA, the copper catalyzed "Click" reaction was used as described in the Click-iT (ThermoFisher, C10640) protocol[93]. Cells were blocked with blocking buffer (2% glycine, 2% BSA, 0.2% gelatin, and 50 mM $NH_4Cl$ in PBS) for 1 h at RT or overnight at 4 °C prior to further staining.

**ssDNA and IF labeling**. To visualize ssDNA, cells were cultured with BrdU either 11 h prior to, and during, CPT treatment, or during CPT treatment only. No difference in overlap quantification was seen between approaches. After cell fixation and the naDNA "Click" reaction, BrdU was then immunolabeled alongside protein with mouse monoclonal anti-BrdU (Abcam, 8039). This approach, without any denaturation of the DNA, has previously been demonstrated as limiting antibody access to, and fluorescent tagging of, the ssDNA only[30]. Proteins of interest were labeled using either direct or indirect labeling with Alexa Fluor 488 and 568 fluorophore-labeled antibodies that have either been previously validated in IF experiments or were cross-validated by us as listed in Supplementary Table 4.

**Super-resolution imaging**. Coverslips with fixed cells were stored for up to 1 week at 4 °C prior to SR imaging. They were then mounted into a microscope microfluidics chamber and SR imaging buffer flowed through. This buffer comprised an oxygen scavenging system (1 mg/mL glucose oxidase (SigmaAldrich, G2133), 0.02 mg/mL catalase (SigmaAldrich, C3155), and 10% glucose (SigmaAldrich, G8270)) and 100 mM mercaptoethylamine (Fisher Scientific, BP2664100) in PBS, pH = 8[94].

All raw SR images were acquired using an in-house custom-built SR microscope based on a Leica DMI 3000 inverted microscope as has been described previously[95]. Briefly, 473 nm (Opto Engine LLC, MBL-473-300 mW), 532 nm (OEM Laser Systems, MLL-III 200 mW), 556 nm (UltraLasers, MGL-FN-556 200 mW), and 640 nm (OEM Laser Systems, MLL-III 150 mW) laser lines were combined using appropriate dichroics and focused onto the back aperture of a HCX PL APO 100X NA = 1.47 TIRF (Leica) objective via a multi-band dichroic (Chroma, zt405/488/532/640/730rpc, UF1C165837). A highly inclined and laminated optical illumination configuration could be achieved by translating the excitation beam laterally across the back aperture of the objective. This helped limit out-of-plane fluorescence and increased power density. Fluorescence emission was collected through the objective and dichroic mirror and focused on an electron multiplying charge coupled device (EM-CCD) camera (Andor iXon+897). Fluorescence signal from AF568 and AF647 could be collected simultaneously using a dual-band bandpass filter (Chroma, CY3/CY5, 59007 m) and split into two channels on the EM-CCD using a dichroic mirror (Semrock, FF660-Di02) in a dual-view cube (Photometrics, DV2). AF488 signal was collected subsequently using a narrow single-band filter (Semrock, FF01-531/40). A 405 nm laser line (Applied Scientific Pro., SL-405 nm-300 mW) was introduced to enhance recovery of dark-state fluorophores when required. AF647-conjugated nascent DNA was used to identify S-phase nuclei (approximately 80% of cells present), and 2000 frames at 33 Hz were acquired for each color.

To correct for geometric offsets and chromatic aberrations caused by the varying diffraction behaviors of different energy emissions, a polynomial morph-type mapping algorithm was applied to the three-color images. The correcting map was generated before each experiment by imaging spatially separated Tetraspecks in each of the three channels (Tetraspecks, 100 nm, Life Technologies, T7279). The localizations of the beads were calculated by independently fitting their diffraction-limited Point Spread Functions (PSF) with Gaussian functions. The localizations could then be matched across the three color channels by fitting a third polynomial function using an IDL (Exelis Visual Information Solutions) custom script. Raw multicolor cell image stacks were then spatially corrected using this polynomial function before SR rendering and analysis. Mapping error distances were calculated by imaging multicolored beads and found to be <10 nm when mapping the blue channel to the red and ~10 nm when mapping the green channel to the red (Supplementary Fig. 5). Fitting uncertainty was also determined to be ~9.5, 9.3, and 17.7 nm in red, blue, and green, respectively.

The camera reading background varies with different microscopic configurations. In brief, with our current illumination and EM gain configurations (as detailed in the Methods section), the camera reading background is ~230, ~170, and ~320 analog-to-digital unit (ADU) in AF647, AF568, and AF488 channels, respectively. Such background as well as other background signal (e.g., out-of-focus fluorescence, auto-fluorescence) is subtracted before localizing individual single-molecule[14]. Besides the already subtracted camera reading background, the background for single-molecule localization is characterized by calculating the standard deviation on a small area of $2 \times 2$ micron$^2$ centered at the minimum intensity pixel of each frame of the image stack, and such background is usually about ~12–16 ADU.

**Super resolution image rendering and analysis**. The ImageJ[96] plugin Quick-PALM[97] was used to analyze and render SR images with point spread function fitting constrained to spots with full-width half-maximum up to 640 nm and signal-to-noise ratios better than 3. Images were rendered with 20 nm pixels and the individual color channels recombined. Individual nuclei were identified based on naDNA signal and masked manually for further analysis.

For an overview of our analyses, see Supplementary Fig. 2. To calculate the degree of colocalization between molecular species, we designed an analysis pipeline that minimized bias and assumptions. Each nucleus was manually outlined to generate an ROI for independent analysis. For each nucleus, an automatic Otsu threshold was applied and the clusters defined for each color. For this analysis, images were analyzed two colors at a time. To generate a baseline of expected random colocalization, the clusters of one color were redistributed within the ROI using a Monte Carlo randomization algorithm and the number and area of overlaps calculated in each simulation[82]. The number/area of overlaps in the real image was then ratioed to the average number/area of overlaps generated from the simulations, thus normalizing for the amount of overlap expected due to random colocalization and the two-dimensional (2D) projection of 3D data. A colocalization ratio of 1 indicated completely random colocalization, while higher numbers indicated association and interaction. Colocalization factors were also calculated for undamaged cells to establish control levels allowing for comparison with CPT and siRNA-treated cells. Number of overlaps was used for MRE11, BRCA1, CtIP, and BLM, whereas area of overlap was used for γH2AX, RPA, RAD51, RAD52, and BRCA2 due to the expected accumulation of these proteins over multiple time points.

To assess the prevalence of colocalization, dependence, or exclusionary relationships between proteins at DSBs, all foci within a nucleus containing both naDNA and at least one of the two proteins stained were identified and quantified manually to determine the percentage of foci with colocalized proteins, as opposed to those positive only for one or the other. The average proportions were calculated and depicted in cumulative bar graphs.

The intercluster distances of different proteins present at the same naDNA foci were calculated in order to assess their spatial relationships. To do this, three-color positive foci were manually examined and the linear distance between the centers-of-mass of the two colors representing the two proteins was determined. These distances were then used to generate a histogram that was fit using either a single or double Gaussian, which was further extrapolated into a 2D likelihood intensity map by fitting perpendicular Gaussians based on the frequency intensity. Double-labeled RAD51 present in cells at repair foci was used to generate a protein–protein distribution map descriptive of closely associated proteins (Supplementary Fig. 1c). We found that this data could be well fit with a Gaussian centered at an intrafoci distance of 135 nm with a full width at half height of 70–80 nm. In cases where protein intrafoci distance histograms could similarly be fit with a Gaussian centered with the range 135 ± 37.5 nm, the relationship was considered proximal. Distances that were approximated with a single Gaussian centered further than 180 nm described proteins occupying the same DSB naDNA but spatially separated. Histograms not easily described by a single Gaussian were well approximated by fitting one Gaussian at 135 nm distance and 75 nm full width at half height while a second Gaussian describing the distal interactions fit using free parameters.

The representative image of nuclei included in this report show both three-color epifluorescence images constructed by producing projections of all the raw data that contributed to the SR image and then adjusting brightness for each color to allow visualization of all three channels. The SR images shown, as well as the various representative foci shown throughout the publication, was generated in single colors using QuickPALM and then combined, cropped, and thresholded to allow for visualization of sub-diffraction foci colocalizations. For display purposes, SR images have been brightness adjusted and smoothed to better show readers the clustered and overlapping signals. Importantly, all quantitative analysis used here was applied to images that had not been manually processed, removing any user bias.

**Comet assays**. To assess the amount of DNA DSBs generated by siRNA treatments, a neutral comet assay was used. Cells were cultured in a six-well plate and treated as outlined above. For the comet assay, these cells were then incubated with 0.4 mL of Trypsin for 10 min. One milliliter of medium was added so that the cells could be suspended and then centrifuged at $1000 \times g$ for 5 min. The cells were then resuspended in 500 μL of fresh PBS and added to prewarmed low-melting point agarose at 37 °C (10 μL of cells to 90 μL of agarose). This suspension was pipetted onto a CometSlide (4250, Trevigen) and spread equally before being allowed to set for 30 min at 4 °C. The slides were then submerged in cold Lysis Buffer (4250, Trevigen) for 30 min on a shaker and then in cold Tris/Borate/EDTA buffer. Electrophoresis was run for 15 min at 21 V in a Mini-Sub Cell GT electrophoresis tray (Bio-Rad). Subsequently, cells were fixed in 70% ethanol and allowed to dry overnight. DNA was stained with Cygreen (GEN-105, ENZO) for 30 min, and slides were imaged on an EVOS fluorescence microscope (AMG) with appropriate filters for green fluorescent protein imaging. Images acquired were processed using the Open Comet software and the olive moment for each group of cells was calculated[98].

**Quantification and statistical analysis**. All experiments were carried out in duplicate or more with repeats performed to generate approximately normal data distributions with N sizes not predetermined. Unequal variances, particularly across temporal series, are expected. For full N values and t tests, see Supplementary Fig. 6 and Supplementary Tables 1–3. Manual ROI selection of nuclei for all quantification, as well as data assessment for pairwise inter-dependency assessment and intrafoci association distribution, was carried out blinded to the drug condition, protein species, and time point before quantification. Statistical analysis was carried out in OriginLab (8.5). All naDNA/protein overlap colocalization factors calculated for controls and CPT-treated cells were tested for significance across time points and against control samples (Student's two-sample t test).

**Code availability**. IDL code used for polynomial morph mapping of channels will be made available upon request. ImageJ plugins for QuickPALM[97] and the randomization calculation[82] can be downloaded from the source publications.

## Data availability

The raw SR data and rendered single-molecule images constitute a sizeable dataset (>10 TB) that cannot be reasonably maintained online. Data will be made available by the corresponding author upon request.

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

## Acknowledgements

We thank members of the Rothenberg laboratory and Nick Cowan for critically reading and commenting on the manuscript. Research in the Rothenberg laboratory is funded by grants from the National Institutes of Health (NIH: R01 GM108119), American Cancer Society (ACS: 130304-RSG-16-241-01-DMC), and the V Foundation for Cancer Research (D2018-020). D.R.W. would like to acknowledge that she is currently supported by a Bruce Stone Fellowship from the La Trobe Institute for Molecular Science.

## Author contributions

D.R.W. and E.R. designed the study and wrote the manuscript with input from all the authors. D.R.W. prepared the samples and performed the super-resolution imaging experiments. W.T.C.L. and D.M.O. prepared the siRNA samples and performed the comet assays. D.R.W. and Y.Y. maintained equipment and performed calibration measurements. D.R.W. analyzed the data using analytical approaches and custom code developed by D.R.W., Y.Y., K.B.-H., S.K., and D.F.

## Additional information

**Competing interests:** The authors declare no competing interests.

