## [Peer Review File · Nature Communications]

Reviewers' comments:

Reviewer #1 (Remarks to the Author):

This manuscript describes the mechanism regulating the recruitment of RAD51 at camptothecin (CPT)-induced one-sided DSB (osDSB), associated with the collapse of stalled replication forks, using super-resolution (SR) microscopy. Exploiting the multi-colour SR analyses of nascent DNA (active replication forks), ssDNA (resected DNA), RAD51, RAD52, BRCA2 and other DNA repair proteins in various genetic backgrounds, they propose a model that the initial assembly of RAD51 is mediated by RAD52, but this role can be compensated by BRCA2. Further, they propose the new role of BRCA1 in later phase of HR.

This paper is potentially interesting, but in this reviewer's view has serious issues to support their conclusions and therefore premature for the publication in Nature Communication.

The main problem with this reviewer is following. The authors propose the involvement of RAD52 in the early recruitment of RAD51, but this model is solely based on the single result that RAD52 arrives at naDNA a little earlier than BRCA2, albeit not strongly, as shown in Figure 2A (at 2 hours). How confident is this earlier arrival of RAD52? Are the sensitivities of these antibodies equivalent? It is difficult to assess, and pictures of naDNA-RAD52-RAD51 co-localisation are not presented. The N values of RAD52 or BRCA2 co-localisation with naDNA at 2 hours, which are currently 17 (Table 1), should also be increased to 40-50, as is the case for RAD51.

Figure 4A shows that the RAD51 recruitment in RAD52 down-regulated cells was normal, which is in contradiction of their model. They explain that BRCA2 plays the compensating role. This notion builds on their observation that, in the absence of RAD52, BRCA2 colocalises strongly with naDNA even at very early time point (Figure 4B). If their model is correct, they should observe the total impairment of RAD51 recruitment upon the down-regulation of both RAD52 and BRCA2 (similarly to Figure 6A, which shows the strong impairment of RAD51-naDNA colocalisation following RAD52/BRCA1 down-regulation). In the meantime, there are alternative explanations for the observed increase of BRCA2 association with naDNA. RAD52 is shown to act against the protection of stalled replication forks, which involves fork reversal mediated by BRCA2 and RAD51 (Mijic, 2017). Hence, the RAD52 down-regulation may actually enhance this protective process, with more stable BRCA2 association with stalled replication forks. Conversely, recent evidence demonstrates that human RAD52 can mediate break-induced DNA replication (BIR) in RAD51 independent manner (Bhowmick, 2016; Sotiriou SK, 2016; Min, 2017). In the light of these findings, one can envisage that, in RAD52 siRNA treated cells, RAD51-BRCA2 mediated HR process compensates the RAD52-mediated BIR.

Figure 4A also shows that the initial RAD51 recruitment was likewise intact upon BRCA2 depletion. This reviewer found the efficiency of BRCA2 down-regulation quite poor (SI Figure 2a), and it is possible that small amount of remaining BRCA2 is enough to recruit RAD51. Nevertheless, this finding is not overly surprising given the previous reports showing the mechanisms that mediate RAD51 recruitment to stalled replication forks independently of BRCA2, such as the MMS22L-TONSL complex and PLK1-dependent RAD51 phosphorylation (Tarsounas et al., 2003; Piwko et al., 2016; Moudry et al., 2016). In line, the description 'it is believed that in higher eukaryote RAD51 mediation is undertaken exclusively by BRCA2' is an exaggeration.

Their observation is also in discrepancy with previous reports that BRCA2 localises at DSB at very early stage (Haas KT et al., NAR 2018; Zhang F et al., 2015, Cell Reports). However, this reviewer understands that the mechanism regulating BRCA2 recruitment to IR-induced DSB and CPT-induced collapsed replication forks could be different. This aspect can be highlighted in the manuscript.

Finally, this reviewer also found no strong evidence supporting the notion that BRCA1 functions in

the later phase of HR. It is widely described that, in addition to the role of BRCA1 in CtIP-mediated resection, BRCA1 also physically binds to PALB2, which in turn recruits the BRCA2-RAD51 complex at sites of DSB. In this way, BRCA1 promotes HR repair. This explains the impaired naDNA co-localisation with BRCA2 and RAD51 upon BRCA1 single depletion (Figure 5E) or BRCA1/RAD52 double depletion (Figure 6), respectively. The increased RAD51 colocalisation in BRCA1 singly depleted cells (Figure 5D) is puzzling, but can be explained by the increased level of DNA damage in the absence of the BRCA1-PALB2-BRCA2-RAD51 pathway, which is flowingly repaired by the RAD52-RAD51 pathway (Figure 6A), as has been proposed previously.

Other points -

Depictions of stalled replication fork, resection and/or the HR repair of collapsed fork (shown in Figs 1A, 4C, 5F and 6C) are confusing. If the broken DNA is the leading strand template, as is often considered to be the case, 5'-ssDNA of template strand cannot serve for RAD51-mediated strand invasion. Rather, the leading strand template should be resected to generate the 3'-end of leading naDNA strand, which can then be coated by RAD51 for HR repair. According to the method section, there was no difference in colocalisation between samples with different BrdU labelling approaches - namely, 11 hours prior to, and during CPT treatment, or during CPT treatment only. Hence, it is likely the BrdU-containing ssDNA is indeed naDNA (mixed with EdU). This is in line with the kinetics of ssDNA colocalising with nsDNA, shown in Figure 1C, which increase over recovery time, peaking at 4 hours.

There are a number of typos (e.g., the name of human proteins should be all capitalised p9, p32; ssDA/RAD51 in p18 should be read ssDNA/RAD51) and missing information in the method section (e.g., the abbreviation of HRR p32; product codes and supplying companies in pages 32, 33).

Reviewer #2 (Remarks to the Author):

In the present study, Whelan DR et al. elegantly demonstrate the spatiotemporal dynamics of homologous recombination (HR) factors in response to replication stress. Because the authors' team has established effective techniques for super-resolution analysis, the data presented in this study are convincing. To the best of our knowledge, this is the first study that demonstrates the spatiotemporal dynamics in HR proteins using super-resolution microscopy. Overall, the remarkable observations have been made, the manuscript is well written and the proposed model is reasonable. However, since this study's conclusions have already been reported by studies using molecular biology techniques, the present study is not entirely novel. In addition, there are some concerns regarding the assays and reagents used, and some controls. Thus, this manuscript could be of interest but further work is required for proving the interpretation and providing novel insights for readers of top journals like Nature Communications.

Major points

1. A major concern in this study is the dose of CPT treatment. It has been shown that low- or high-dose CPT treatment generates different types of DNA damage, i.e., either seDSB or chicken-foot structure via replication-fork reversal without forming a DSB (Arnab Ray Chaudhuri et al., NSMB, 2012). In the present study, 100 nM CPT was used and the authors state that the treatment induced single-ended DSB (seDSB). Please also note that one-sided DSB is not a commonly used term in the field. The authors should call it single-ended DSB (seDSB) or one-ended DSB. The authors should also verify whether the observed DNA damage response was associated with the formation of seDSB. If fork reversal is caused as a response to 100 uM CPT, these foci may be dependent on the DSB ends of chicken-fork structures without generating an seDSB. Because BRCA2 also plays a role in the repair of replication collapse (Schlachter K, Cell, 2011), the interpretations of this study may need correction.

In addition, for further confirmation of the findings of this study, PARP inhibitor or other reagents

should be tested.

2. If RPA/RAD51 foci formation is resection-dependent, these foci will be diminished by CtIP depletion. The authors should test this.

3. Within Figure 1C, the authors should show raw images of each protein for each condition (both a cell and enlarged foci) in a supplemental figure. In general, the shape of foci changes with time. This should be clarified and stated, e.g., the shapes of the RAD51 foci do not look similar in Figure 2B. Did the authors not observe clustered foci as described in a previous study (Natale et al., Nat Comm., 2017)?

4. Within Figure 1C, why does the number of RPA foci go up again at 8 h? This raises a critical concern regarding the interpretation of RPA foci, because the authors may detect RPA at D-loop structures as well as RPA at resected-DSB. RPA may bind to ssDNA at D-loops and form foci, even though such RPA is unlikely to be identified using conventional microscopy. However, the single molecule analysis may pick up small numbers of RPA at D-loops. If so, the interpretation of Figure 2B may need to be amended. Even though the authors state that RPA and RAD51 share binding onto resected-ssDNA, this may not be the case. The authors should clarify this point, or the possibility should at least be discussed so that definitive statements are avoided in the manuscript.

5. The rationale that RAD52 is required for early RAD51 recruitment is unclear. For consolidating this idea, the authors should show no RAD51 recruitment in RAD52/BRCA2 double-knockdown cells. Because double-knockdown is lethal, the experiment may be infeasible. However, since RAD52/BRCA1 double-knockdown was examined, it is worthwhile to test RAD52/BRCA2 knockdown as well. In the manuscript, the authors state that 'Combined RAD52/BRCA1 deficiencies have been shown to lead to synthetic lethality similar to that seen in combined RAD52/BRCA2 deficient cells'. Thus, testing RAD52/BRCA2 knockdown at earlier time points may be feasible. If the experiment is infeasible, the authors should explicitly state why the results support the conclusion that RAD52 is required for RAD51 recruitment at an early phase, and is not sustained at ssDNA in the absence of BRCA2.

6. The role of BRCA1 for RAD51 filament formation by the recruitment of BRCA2 via PALB2 has been demonstrated in several studies (Buisson et al., 2017; Orthwein et al., 2015; Zhang et al., 2009). Therefore, a major finding in Figures 5 and 6 is that RAD52 promotes BRCA2 in the absence of BRCA1. In Figure 5E, the authors demonstrate a reduction in BRCA2 recruitment in BRCA1-depleted cells, but importantly, BRCA2 recruitment is not completely abolished. In contrast to results for BRCA1 knockdown alone, the authors show no BRCA2 recruitment in RAD52/BRCA1 double-knockdown cells. Even though this is an important finding, the authors unfortunately used different time points. The authors should test BRCA2 recruitment in control, BRCA1, RAD52 and RAD52/BRCA1 double-knockdown cells at the same time point. If RAD52 has a role in promoting BRCA2, an arrow from RAD52 to BRCA2 should be shown in Figure 6C (ii) for WT. This will provide novel insights into the mechanism of RAD51 loading via BRCA1/2 in response to replication stress.

Minor points

7. In multi-colour staining, each colour may show a one- or half-pixel shift. The authors should provide information regarding pixel shift for each colour. Pixel shift should be examined using multi-colour TetraSpeck™ beads.

8. In Figure 3A, the purpose of the small panels is unclear, although the authors state that they are representative images. For example, in the middle panel of Figure 3A, MRE11 foci do not overlap with ssDNA. The authors should clarify the intended meaning of this point.

Reviewer #3 (Remarks to the Author):

Rothenberg and coworkers use Super-resolution fluorescence imaging to observe the temporal dynamics of 4 proteins associated with DNA repair through homologous recombination. They use colocalization of different protein clusters/ puncta on the damage site to establish the order of arrival, cross talk and epistasis between the proteins BRCA1 and 2 and RAD51 and 52. While this study is very informative for the field of DNA damage and repair, this reviewer does not see any novelty in the approach. Further, the study relies on STORM imaging, yet there is no information on critical SR imaging aspects such as localization precision, and most importantly channel dependent and field dependent registration error have been completely ignored. There is no information on background fluorescence, # of localizations per cluster etc. It is clear that any claims based on co-localization of fluorescence spots in multiple channels must be backed by rigorous controls, both biological and through optics/ simulations.

This reviewer finds the manuscript lacking these crucial pieces of information. Based on these observations, this reviewer does not recommend the publication of this manuscript in Nat. Comm. in its current form and further recommends that the authors target their manuscript to a more specialized journal.

Point-by-point response to the reviewers' comments for manuscript No. NCOMMS-18-01634-T

Spatiotemporal Dynamics of Homologous Recombination Repair at Single Collapsed Replication Forks

Donna R. Whelan, Wei Ting C. Lee, Yandong Yin, Dylan M. Ofri, Keria Bermudez-Hernandez, Sarah Keegan, David Fenyo & Eli Rothenberg.

Black text denotes reviewer comments. Blue text denotes point-by-point responses. Red denotes changes to the manuscript and SI.

Reviewer #1 (Remarks to the Author):

This manuscript describes the mechanism regulating the recruitment of RAD51 at camptothecin (CPT)-induced one-sided DSB (osDSB), associated with the collapse of stalled replication forks, using super-resolution (SR) microscopy. Exploiting the multi-colour SR analyses of nascent DNA (active replication forks), ssDNA (resected DNA), RAD51, RAD52, BRCA2 and other DNA repair proteins in various genetic backgrounds, they propose a model that the initial assembly of RAD51 is mediated by RAD52, but this role can be compensated by BRCA2. Further, they propose the new role of BRCA1 in later phase of HR.

This paper is potentially interesting, but in this reviewer's view has serious issues to support their conclusions and therefore premature for the publication in Nature Communication. The main problem with this reviewer is following.

(1) The authors propose the involvement of RAD52 in the early recruitment of RAD51, but this model is solely based on the single result that RAD52 arrives at naDNA a little earlier than BRCA2, albeit not strongly, as shown in Figure 2A (at 2 hours). How confident is this earlier arrival of RAD52? Are the sensitivities of these antibodies equivalent? It is difficult to assess, and pictures of naDNA-RAD52-RAD51

co-localisation are not presented. The N values of RAD52 or BRCA2 co-localisation with naDNA at 2 hours, which are currently 17 (Table 1), should also be increased to 40-50, as is the case for RAD51.

In response to this comment we have generated and analysed additional cells to reach the indicated N values and confirmed the statistically significant increase in RAD52 colocalization with naDNA repair foci at 2 hours after CPT treatment, as well as the lack of detectable BRCA2 association at this time (SI Table 1, marked in red). This is also updated in the data shown in Figures 1C, 2B-C, 5B, 5D, 6E). We have also redesigned Figure 2A-C to better present this data which previously used levels of protein at 1 hour post-CPT to normalize to zero. We believe the improved presentation better shows the increase in RAD52 from un-colocalized control levels. We have also included some explanatory text in the SI discussion:

“Colocalization coefficients calculated for undamaged control cells were routinely determined to be above random due to HR protein associations with endogenously stressed and broken RFs, and also because of their transient binding to unwound DNA. For example in Figure 2A, control levels RAD51 overlap with naDNA were significantly higher than random (Colocalization of ~1.5 compared to 1 when random) while RAD52 was slightly higher. Accordingly, we determined changes in colocalization comparative to control levels – for example, after 1 hour of damage, both RAD51 and RAD52 were detected at control levels, before significantly accumulating in the following hour of repair and remained prevalently colocalized with the naDNA foci until more than 12 hours of recovery (Figure 2A-B). In only a handful of cases were specific proteins detected as actively excluded from undamaged replication foci (for example BRCA2, Figure 2C). Controls levels are thus crucial in assessing changes in associations, whereas the randomized simulation approach is more important in normalizing for differences in cell size, cluster size, cluster number, etc. Effectively, we are normalizing each cell’s overlaps to its own eccentricities, allowing for sensitive detection of changes across the cell population upon DDR. We have previously published an extensive discussion of this method, as well as the calculations and ImageJ plugin¹.

We have also amended the figure caption to provide further detail:

“RAD52 colocalizes with RAD51 at damage sites prior to BRCA2 colocalization.

(A-C) Kinetics of (A) RAD51, (B) RAD52, and (C) BRCA2 colocalization throughout repair (over 16 hours of recovery). For complete N values see SI Table 1. Values were calculated using the detailed Monte Carlo randomization method as described in Methods, SI Figure 1, and SI Discussion. This allowed the detected number/area of colocalization for each cell to be normalized to the predicted number/area of colocalization in a random simulation of the same cellular image. As plotted here: 1 indicates random overlap (shown as red dashed line) whereas 2 indicates double the number/area of overlaps as expected based on the randomized model. Colocalization in undamaged control cells also shown. Error bars represent s.e.m. Student's t-test for significance between control and damage levels.”

Our initial hypothesis that RAD52 is involved in early RAD51 recruitment was based on the RAD52 presence at 2 hours, prior to BRCA2 arrival. We considered this in contrast to the more generally accepted belief that BRCA2 recruits RAD51 and tested our theory throughout this paper using the knockdowns and colocalization imaging. To further strengthen these points, we have now also incorporated pairwise analysis into Figure 2D showing strong colocalization of RAD51 and RAD52 at repair foci after 2 hours. Similar analysis of RAD51 and BRCA2 was prohibited by the lack of BRCA2 present at this time (Figure 2C). We also include representative images of naDNA/RAD51/RAD52 and naDNA/RAD51/BRCA2 as these time points with magnified foci showing the temporal progression. We have also discussed this in the text at line 143:

“These unique RAD51/RAD52 and RAD51/BRCA2 localization trends at 2 and 4 hours of recovery are readily observed in the SR images of cell nuclei and individual seDSB foci as shown in Figure 2D-H; At 2 hours more than 60% of the foci contain both RAD51 and RAD52 (Figure 2D-E), while co-analysis of BRCA2 and RAD51 shows a lack of BRCA2 at the foci despite the presence of RAD51 (Figure 2E). By 4 hours, BRCA2 could be observed colocalized with RAD51 at repair foci, however this remained in contrast to RAD52/RAD51 association which could be observed at both 2 and 4 hours recovery (Figure 2F-H).”

Overall we have tried to minimise using excessive images in the manuscript due to the difficulty of printing large SR images at 300 dpi (single foci can easily end up smaller than 1/300th of an inch) and the fact that our data present a highly detailed description of foci organization that differs from conventional confocal overlapped ‘foci’. We have included an extensive discussion of these differences in the new SI

Discussion, sub-headed “Complications arising from SR imaging”. To minimize the use of the SR pseudo-images themselves, we rely on computational approaches and analysis to detect hundreds of small-scale, but genuine, overlaps per cell. This also allows for lower N numbers – for each cell imaged we are able to differentiate and quantify hundreds or thousands of replication and protein foci, usually with dozens to hundreds of overlaps, we have emphasised this in the introduction at line 60:

“Here we overcome these limitations by using multicolor super-resolution imaging, which provides a ten-fold improvement in spatial resolution as well as single molecule sensitivity^{31,32}. In combination with assays that specifically label nascent DNA (naDNA), ssDNA, and proteins associated with repair foci, we could detect and examine individual seDSB sites in vivo, routinely detecting several hundred individual naDNA and protein foci in a single cell image with tens to hundreds of overlaps.”

(2) Figure 4A shows that the RAD51 recruitment in RAD52 down-regulated cells was normal, which is in contradiction of their model. They explain that BRCA2 plays the compensating role. This notion builds on their observation that, in the absence of RAD52, BRCA2 colocalises strongly with naDNA even at very early time point (Figure 4B). If their model is correct, they should observe the total impairment of RAD51 recruitment upon the down-regulation of both RAD52 and BRCA2 (similarly to Figure 6A, which shows the strong impairment of RAD51-naDNA colocalisation following RAD52/BRCA1 down-regulation).

The reviewer raises a good point and we have spent significant time over the past few months attempting to optimize and image double knock-down siRAD52/siBRCA2 cells. As Reviewer 2 predicted, the combined synthetic lethality of these two deficiencies prohibits any meaningful imaging, as siRNA treatment resulted in rapid cell death. Similar, although slightly less severe results, were seen in the siRAD52/siBRCA1 cells which could not be imaged for more than a couple of hours after CPT treatment and also showed a much lower percentage of cells in S-phase (based on EdU label). While we could not measure RAD51 recruitment in siRAD52/siBRCA2 cells, we could confirm the synthetic lethality of these deficiencies which agrees well with previous studies and our new conclusions. We have added the following to line 314 to address this:

“Attempts to prepare double-depleted siRAD52/siBRCA2 cells similarly demonstrated this synthetic lethality with most cells succumbing to apoptosis during the transfection process.”

(3) In the meantime, there are alternative explanations for the observed increase of BRCA2 association with naDNA. RAD52 is shown to act against the protection of stalled replication forks, which involves fork reversal mediated by BRCA2 and RAD51 (Mijic, 2017). Hence, the RAD52 down-regulation may actually enhance this protective process, with more stable BRCA2 association with stalled replication forks. Conversely, recent evidence demonstrates that human RAD52 can mediate break-induced DNA replication (BIR) in RAD51 independent manner (Sotiriou SK, 2016; Min, 2017). In the light of these findings, one can envisage that, in RAD52 siRNA treated cells, RAD51-BRCA2 mediated HR process compensates the RAD52-mediated BIR.

In response, we have included comet assays that show significant induction of DSBs in CPT-treated WT cells, thus confirming that although CPT can induce RF stalling, we are also generating DSBs at RFs (SI Figure 2 and 4). Importantly, the naDNA foci we visualize and use as markers for DSBs also colocalize strongly with typical HR markers including MRE11, CtIP, BRCA and RPA (Figure 1C). These correlate and colocalize with RAD51 strongly demonstrating that these foci harbor DSBs and that we are measuring resection associated events and HR at seDSBs rather than protections/repair events at stalled forks. As to the possibility that in the RAD52 knockdown cells, BRCA2 acts to stabilize stalled forks, this is contrary to the similar numbers of DSBs induced by CPT in control and siRAD52 cells (SI Figure 4 comet assays). The query of RAD52 mediated BIR is quite interesting, however we know no assay which would differentiate individual break sites undergoing BIR vs other HR repair. Moreover, we do not observe significant RAD52 association with repair foci in the absence of RAD51 at any point during our experiments making RAD52-mediated, RAD51 independent repair unlikely.

We have included these points as part of a new SI Discussion, sub-headed ‘Other repair pathways and damage motifs’:

“Despite evidence that HR is the predominant repair pathways for seDSBs in S-phase cells, we also considered the possibility that we were observed RAD52-dependent/RAD51-independent break induced replication (BIR) as has been described previously^{22,23}. Discerning BIR from canonical HR remains extremely challenging, however the persistent of RAD51 at RAD52-positive repair foci made it unlikely that we were detecting such a RAD51-independent pathways.”

“Our surprising results regarding RAD52’s role in HR as a RAD51-mediator also prompted us to consider previous studies which identified RAD52 as an antagonist of stalled RF stability²⁴ whereas BRCA2 is a known mediator of RF protection²⁵. We thus considered the possibility that the BRCA2 association we detected in siRAD52-treated cell (Figure 5B) was due to its recruitment to unbroken RFs in the absence of RAD52-mediated degradation. However, this did not explain BRCA2’s colocalization with DSB-specific proteins such as RPA, RAD51 and BRCA1, nor did it account for BRCA2’s persistence over more than eight hours of recovery. Furthermore, BRCA2-dependent protection of RFs should have resulted in a decrease in the number of breaks generated, which we could not detect via comet assay (SI Figure 4).”

(4) Figure 4A also shows that the initial RAD51 recruitment was likewise intact upon BRCA2 depletion. This reviewer found the efficiency of BRCA2 down-regulation quite poor (SI Figure 2a), and it is possible that small amount of remaining BRCA2 is enough to recruit RAD51. Nevertheless, this finding is not overly surprising given the previous reports showing the mechanisms that mediate RAD51 recruitment to stalled replication forks independently of BRCA2, such as the MMS22L-TONSL complex and PLK1-dependent RAD51 phosphorylation (Tarsounas et al., 2003; Piwko et al., 2016; Moudry et al., 2016). In line, the description ‘it is believed that in higher eukaryote RAD51 mediation is undertaken exclusively by BRCA2’ is an exaggeration.

The reviewer makes a good point regarding previous work showing RAD51 at replication forks independent of BRCA2 mediation however we strongly believe that the majority of people in the field consider BRCA2 to be the principal RAD51 mediator when it comes to endogenous HR, something we show is incorrect in our manuscript. However, we acknowledge that our use of ‘exclusively’ might be misplaced and would

like to acknowledge these studies as they support our views on the complexity of this process. As such as have added (line 151 and 246, respectively):

“These observations were of particular interest because, while the Rad52 protein in unicellular eukaryotes is critical for Rad51 function, it is believed that in higher eukaryotes BRCA2 is the principal facilitator of ssDNA/RAD51 nucleofilament formation and function².”

“This was in agreement with previous studies that describe BRCA2-independent RAD51 association with stalled replication forks^{3,4}. However, recruitment of RAD51 to seDSBs 2 hours following CPT treatment was also detected at similar levels in WT and BRCA2-depleted cells (Figure 5A), demonstrating that the initial RAD51 recruitment to both stalled and broken forks is BRCA2-independent.”

Regarding the level of BRCA2 depletion: in the WT CPT damaged cells, we considered the possibility that we were unable to detect early BRCA2 activity due to the transient nature of the interaction, however the strong presence of BRCA2 in siRAD52 cells demonstrates that when it is recruited to the foci it is detected quite well. Thus despite evidence that some BRCA2 remains in the cell upon knockdown, we ascribe RAD51 recruitment to RAD52 which colocalizes at these early time points in much the same pattern as BRCA2 does in siRAD52 cells.

(5) Their observation is also in discrepancy with previous reports that BRCA2 localises at DSB at very early stage (Haas KT et al., NAR 2018; Zhang F et al., 2015, Cell Reports). However, this reviewer understands that the mechanism regulating BRCA2 recruitment to IR-induced DSB and CPT-induced collapsed replication forks could be different. This aspect can be highlighted in the manuscript.

We thank the reviewer for their appreciation of this novelty of our work and are happy to highlight these reports and our explanation within the text on line 164.

“Elucidation of delayed BRCA2 recruitment contrasts other studies that have detected its recruitment to DSBs within 30 seconds⁵ or the first half hour following damage⁶, however previous observations of sub-minute protein recruitment have generally relied on massive, heterogeneous damage induced using laser microirradiation in order to achieve high temporal resolution but also confounding the damage response pathways in play and limiting colocalization to confocal

measurements^{5,7-9}. Similarly, two-ended DSBs induced by ionizing radiation are not analogous to the HR-repaired single-ended DSBs such as those we examine here and those that are predominantly caused by endogenous stress¹⁰, potentially resulting in different repair pathways. The variance in data likely caused by the confounding variables introduced by laser and ionizing microirradiation were recently demonstrated in a metadata analysis of protein recruitment to damage sites which showed significant differences between studies¹¹.”

(6) Finally, this reviewer also found no strong evidence supporting the notion that BRCA1 functions in the later phase of HR. It is widely described that, in addition to the role of BRCA1 in CtIP-mediated resection, BRCA1 also physically binds to PALB2, which in turn recruits the BRCA2-RAD51 complex at sites of DSB. In this way, BRCA1 promotes HR repair. This explains the impaired ssDNA co-localisation with BRCA2 and RAD51 upon BRCA1 single depletion (Figure 5E) or BRCA1/RAD52 double depletion (Figure 6), respectively. The increased RAD51 colocalisation in BRCA1 singly depleted cells (Figure 5D) is puzzling, but can be explained by the increased level of DNA damage in the absence of the BRCA1-PALB2-BRCA2-RAD51 pathway, which is slowly repaired by the RAD52-RAD51 pathway (Figure 6A), as has been proposed previously.

We are unsure whether the reviewer is questioning our conclusion that BRCA1 is required for later BRCA2-facilitated HR or questioning this conclusion's novelty. In the case of the former, we believe our siBRCA1 and siBRCA1/siRAD52 experiments clearly show two roles for BRCA1 – firstly in mediating resection, however BRCA1 appear to be dispensable for this as evidenced by delayed but similar levels of RPA association, and secondly in enabling BRCA2 mediation of ssDNA/RAD51 functionality during late HR. This appears to be the more critical of the two BRCA1 functions as the siBRCA1 demonstrates high inhibition of BRCA2 association (Figure 6E) coincident with complete abrogation of RAD51 colocalization at 12 hours (Figure 6D). This further results in an accumulation of DSBs (SI Figure 4F). We believe these observations are extremely novel with regards to endogenous HR at low damage levels because we show that at these levels BRCA1 is dispensable for resection but critical for homology search – such temporal insights have not previously been

possible *in vivo*. While there is ample literature on the many possible roles of BRCA1 this is the strongest evidence of its importance in an endogenous environment. Furthermore, we believe the increased RAD51 levels detected 8 hours after damage (Figure 6D) align with the higher levels of RPA observed at 4 hours (Figure 6C) and are due to delayed resection. We acknowledge that this could be due to an increase in damage sites due to the absence of the BRCA1-PALB2-BRCA2-RAD51 pathway, however we find this a less convincing hypothesis due to the expected loss of RAD51 association coincident with BRCA1 deficiency. We have amended the text to explain these points in more depth, acknowledge more of the current literature, and to consider alternate interpretation (Line 274):

“ To date, multiple roles have been proposed for BRCA1 in HR repair including initial signaling and resection mediation, as well as a potential role upstream of BRCA2 functionality via PALB2 (partner and localizer of BRCA2) binding¹²⁻¹⁵. However the *in vivo* functions of BRCA1 and the relative importance in repair of these mechanisms, have been a source of contention for several decades¹⁶⁻¹⁸. Our detection of BRCA1 at repair foci throughout the HR process (Figure 1C) demonstrates the likelihood of multiple roles, including beyond initial DDR signaling and resection, and possibly related to RAD51/BRCA2 interaction and homology search. To resolve the multiple roles of BRCA1 *in vivo* and to assess their importance, we imaged the localization kinetics of repair proteins in BRCA1-depleted cells. We found that BRCA1 knockdown produced inhibition of early HR by diminishing MRE11 association and blocking CtIP recruitment (Figure 6A-B). This led to a significant delay in resection, with less RPA recruitment detected in siBRCA1 cells compared to in WT cells immediately following damage (Figure 6C). However, RPA association with repair foci reached WT damaged cell levels after a two hour delay (i.e., after 4 hours of recovery from CPT) (Figure 6C)¹⁹. Surprisingly, RAD51 levels were also comparable to those in WT cells, albeit also delayed, reaching a peak at 8 hours instead of at 2-4 hours (Figure 6D). We believe that this result demonstrates that resection was at least partially successful in BRCA1 depleted cells, allowing initial ssDNA/RAD51 nucleofilament formation to proceed by 8 hours. While it is possible that the loss of other protective and reparative BRCA1-dependent pathways induced higher levels of damage (as detected via comet assay, SI Figure 4F), delayed recruitment of RAD51 in the absence of BRCA1 does not seem to indicate this. Furthermore, BRCA2 association - while not fully diminished at 8 hours of recovery - did not reach WT levels at any time point, and RAD51

association also decreased more quickly than expected based on the WT spatiotemporal map (Figure 6D-E). Moreover, despite successful resection and RAD51 loading, BRCA1-depleted cells displayed persistent damage similar to that found in BRCA2-depleted cells (SI Figure 4). These observations are in agreement with the documented genomic instability characteristic of BRCA1 and BRCA2 deficient cells and of mutant disease phenotypes²⁰. We infer that the enhanced DNA damage detected in BRCA1-depleted cells is primarily a consequence of downstream defective BRCA2-mediated ssDNA/RAD51 interaction (Figure 6F).”

And on line 363:

“Surprisingly, although we detect BRCA1 at DSBs during resection and there is ample literature demonstrating its resection mediation abilities^{13,15,21}, by examining BRCA1 depleted cells at stress levels lower than most previous DDR studies, we detect a delay, but no comprehensive deficiency, in resection because both RPA and RAD51 coating of the resected ssDNA reach levels similar to those detected in WT damaged cells. This is further explained by recent studies that identified CtIP-BRCA1 interaction as non-essential for resection but capable of speeding it up^{22,23}. At higher damage levels, it is possible that resection cannot similarly be delayed without further consequences due to limitations on the total amount of repair proteins present and the threshold for damage amount and duration.”

Other points

Depictions of stalled replication fork, resection and/or the HR repair of collapsed fork (shown in Figs 1A, 4C, 5F and 6C) are confusing. If the broken DNA is the leading strand template, as is often considered to be the case, 5'-ssDNA of template strand cannot serve for RAD51-mediated strand invasion. Rather, the leading strand template should be resected to generate the 3'-end of leading naDNA strand, which can then be coated by RAD51 for HR repair. According to the method section, there was no difference in colocalisation between samples with different BrdU labelling approaches - namely, 11 hours prior to, and during CPT treatment, or during CPT treatment only. Hence, it is likely the BrdU-containing ssDNA is indeed naDNA (mixed with EdU). This is in line with the kinetics of ssDNA colocalising with nsDNA, shown in Figure 1C, which increase over recovery time, peaking at 4 hours.

We thank the reviewer for highlighting this oversight and agree with them regarding the BrdU and have remedied the figures accordingly (See Figures 1A, 5C, 6F, and 7C).

There are a number of typos (e.g., the name of human proteins should be all capitalised p9, p32; ssDA/RAD51 in p18 should be read ssDNA/RAD51) and missing information in the method section (e.g., the abbreviation of HRR p32; product codes and supplying companies in pages 32, 33).

We have corrected these oversights.

Reviewer #2 (Remarks to the Author);

In the present study, Whelan DR et al. elegantly demonstrate the spatiotemporal dynamics of homologous recombination (HR) factors in response to replication stress. Because the authors' team has established effective techniques for super-resolution analysis, the data presented in this study are convincing. To the best of our knowledge, this is the first study that demonstrates the spatiotemporal dynamics in HR proteins using super-resolution microscopy. Overall, the remarkable observations have been made, the manuscript is well written and the proposed model is reasonable. However, since this study's conclusions have already been reported by studies using molecular biology techniques, the present study is not entirely novel. In addition, there are some concerns regarding the assays and reagents used, and some controls. Thus, this manuscript could be of interest but further work is required for proving the interpretation and providing novel insights for readers of top journals like Nature Communications.

We thank the reviewer for their positive response and acknowledgement of super-resolution microscopy as a novel, powerful and beneficial technique when it comes to DNA damage response research. We have tried to stress this future potential within the manuscript and appreciate that this reviewer agrees with this. Importantly, we considered our observations inherently novel because our study examines DNA damage in a pseudo-endogenous manner at very low induction rates – something

conventional biochemical and confocal studies have not been able to achieve. In this way we discover that some hypothesized mechanisms and pathways within the cell are indeed present and/or critical, but that others are not. We also specifically uncover the redundancy and relationship between RAD52 and BRCA2 which had not previously been known. By addressing the reviewer's comments and incorporating these points and further data into the manuscript we strongly believe that our paper is worthy of publication in Nature Communications.

Major points:

(1) A major concern in this study is the dose of CPT treatment. It has been shown that low- or high-dose CPT treatment generates different types of DNA damage, i.e., either seDSB or chicken-foot structure via replication-fork reversal without forming a DSB (Arnab Ray Chaudhuri et al., NSMB, 2012). In the present study, 100 nM CPT was used and the authors state that the treatment induced single-ended DSB (seDSB). Please also note that one-sided DSB is not a commonly used term in the field. The authors should call it single-ended DSB (seDSB) or one-ended DSB. The authors should also verify whether the observed DNA damage response was associated with the formation of seDSB. If fork reversal is caused as a response to 100 uM CPT, these foci may be dependent on the DSB ends of chicken-fork structures without generating an seDSB. Because BRCA2 also plays a role in the repair of replication collapse (Schlacher K, Cell, 2011), the interpretations of this study may need correction. In addition, for further confirmation of the findings of this study, PARP inhibitor or other reagents should be tested.

The reviewer raises an important point and one that the research field will need to work to overcome in the future: differentiating RF breaks and stalls *in vivo*. This task proves to be especially challenging due to the high degree of overlap between the repair pathways of the two different stress products. We have endeavored to demonstrate that we are inducing a significant amount of DSBs by including comet assay data (SI Figures 2 and 4) showing a significant increase in CPT-treated cells. Furthermore this data shows that upon Veliparib treatment, even more DSBs are generated, thus providing evidence that CPT does indeed produce stalled but

unbroken RFs which are then protected in a PARP-dependent fashion. Details of this experiment are included in Methods, line 774:

“To inhibit PARP, cells were cultured in the presence of 100 μ M Veliparib (Santa Cruz, 202901) for 24 hours prior to recovery from CPT treatment. To do this, cells seeded on coverslips were administered Veliparib during the final seven hours of serum starvation as well as during release in complete medium and in the CPT/EdU medium. Following CPT treatment, cells were recovered for up to 16 hours in medium free of both CPT and Veliparib.”

Most importantly, we include hydroxyurea-treated cell data to show that RF stalling is quickly repaired compared to the canonical HR process of homology search and repair at seDSBs. Detection of HR/fork protection proteins including BRCA1, RPA, RAD51 and RAD52 are all measured immediately following HU treatment and 90 minutes after the HU is removed, showing no significant association remaining (SI Figure 2). Thus, by treating our cells with CPT for only an hour and then giving them a further hour to recover we believe that the more transient stalled/regressed RF damage sites will have been repaired and so will not confound our observations.

A further point worth considering is the general acceptance of the DDR community of microirradiation as a method of choice for DSB induction, despite the fact that it is known to induce radical oxygen species and indiscriminate damage to proteins and lipids within laser damage vicinity. We do not consider the potential impact of any CPT-induced stalled/regressed forks to be as confounding as this clustered and heterogeneous damage and therefore see our combined CPT-damage/SR imaging approach as imperfect but an improvement over current methods.

To clarify these points we have included the outlined data as SI Figure 2 and 4 and the following text on line 88:

“To assess the generation of seDSBs we analyzed control and CPT-treated cells using a comet assay (SI Figure 2B) and confirmed DSB induction. We ensured that the repair foci we monitor are indeed seDSB, and not stalled RFs²⁴⁻²⁷ by treating S phase cells with a mild dose of hydroxyurea (HU) -- causing RF stalling but not seDSB²⁸ -- and monitored the time course for association of key proteins thought to be involved in both stalled RF rescue and HR (RPA, BRCA2, RAD51 and RAD52). These showed that after 90 minutes recovery from the HU treatment, RF colocalization of these proteins had returned to control levels (SI Figure 2C). We therefore focused

our experiments on cells that were allowed to recover for at least one hour before imaging, as repair foci in these cell will only constitute seDSBs, since any RF stalling events would already be resolved.”

We have also more extensively commented on this in a new SI Discussion, particularly in a subsection titled “Other repair pathways and damage motifs.” We have also changed all references to ‘one-sided DSBs’ to ‘single-ended DSBs’.

(2) If RPA/RAD51 foci formation is resection-dependent, these foci will be diminished by CtIP depletion. The authors should test this.

RPA/RAD51 foci formation is routinely used to detect and quantify resection and so we are confident of this measure. We have further confirmed this using Mirin, an MRE11 inhibitor which completely abrogated colocalization. We have provided this detail and some literature examples on line 107:

“Colocalization in damaged cells was then further assessed by comparison with colocalization levels in control cells with RPA or RAD51 foci formation used to monitor resection projection²⁹⁻³². To demonstrate the efficacy of area of overlap of RPA association as a good measure of resection we treated cells with Mirin, an MRE11 nuclease inhibitor, and detected almost complete abrogation of RPA association with the damage foci (SI Figure 2A).”

(3) Within Figure 1C, the authors should show raw images of each protein for each condition (both a cell and enlarged foci) in a supplemental figure. In general, the shape of foci changes with time. This should be clarified and stated, e.g., the shapes of the RAD51 foci do not look similar in Figure 2B. Did the authors not observe clustered foci as described in a previous study (Natale et al., Nat Comm., 2017)?

The reviewer raises some interesting points, some of which we have addressed previously in other published studies (Baranes-Bachar, K. et al. Mol Cell, 2018, 10.1016/j.molcel.2018.02.002; Young, LM. et al. Cell Rep, 2015, 10.1016/j.celrep.2015.09.017; Reid, DA. Et al. PNAS, 2015, 10.1073/pnas.1420115112). The current data set contains more than 5000 cells with hundreds of clusters present in three channels in each. Each single color cell image

comprises hundreds of thousands of molecule localizations that cannot easily be structurally assessed using the same methods as confocal studies. It is also unclear how this information would immediately complement our primarily temporal observations and crosstalk conclusions.

Nonetheless, we have also included several more whole cell images in Figure 2, focusing on the most interesting protein pairs and contrasting specific time points. We do not see the benefit of including whole cells or 'raw images' in the SI and collation of these images and preparation for print would again be arduous. Our diffraction limited images are collected in widefield and are not particularly telling except for in contrast to the SR images. The SR images themselves are arbitrarily mapped using the 'raw' list of calculated coordinates. Providing a 'true' image is particularly difficult because they are routinely rendered at >20000 pixels wide and thus requiring a high degree of foci dilation and brightening to be visualized on conventional screens and, more so, in print. We strongly believe that the analysis, which is automated and not susceptible to the biases of choosing single cell images, is a far stronger representation of our research. We present these points in a new discussion in the SI specifically focusing on the differences between confocal and SR imaging and the difficulties in data analysis and presentation that these differences present:

While we acknowledge that significant chromatin restructuring is expected to occur during the 16 hours of repair, by labelling the nascent DNA we effectively labelled the break site meaning this label moves alongside the proteins of interest and that more general chromatin rearrangements cannot largely affect our data. We did not notice any significant trends in foci clustering or changing size, with the exception of naDNA foci becoming slightly more diffuse at 12 and 16 hours. These observations do not contradict previous work focused on causing clustered damage and therefore detecting it as such, nor with the work published recently by Natale et al. This research generated two-ended DSBs and so was likely assessing a different DNA damage response. To further demonstrate the lack of clustering in our data we have added an SI figure (SI Figure 5) showing that the raw number of naDNA/RAD51 foci per cell, does not decrease over time, as would be expected if the foci were merging together. We address this in the new SI Discussion:

“A further discrepancy between our data and previous studies is the lack of clustering of repair foci. While a large degree of heterogeneity in both individual foci

structure and overall foci distribution within a cell was observed during our experiments (For examples see Figures 1B, 2E-H, 4A-B), no obvious overall reorganization of DNA repair foci was witnessed. Quantification of the total number of naDNA/RAD51 foci per cell also failed to reveal the expected sharp reduction in foci count predicted by the repair factory hypothesis (SI Figure 5). We suggest that this is due to the low-level, pseudo-endogenous damage that we have examined. Large chromatin rearrangements have previously been seen in studies causing clustered damage or at least generating two-ended DSBs. Here we have only induced a handful of seDSBs and, in doing so, have not triggered repair pathways which make use of repair foci clustering.”

(4) Within Figure 1C, why does the number of RPA foci go up again at 8 h? This raises a critical concern regarding the interpretation of RPA foci, because the authors may detect RPA at D-loop structures as well as RPA at resected-DSB. RPA may bind to ssDNA at D-loops and form foci, even though such RPA is unlikely to be identified using conventional microscopy. However, the single molecule analysis may pick up small numbers of RPA at D-loops. If so, the interpretation of Figure 2B may need to be amended. Even though the authors state that RPA and RAD51 share binding onto resected-ssDNA, this may not be the case. The authors should clarify this point, or the possibility should at least be discussed so that definitive statements are avoided in the manuscript.

To address this issue we calculated changing overlap by area for RPA throughout. Because of this, the small amount of RPA loading at D-loops could only ever have a small impact on our calculations. We are unsure of the exact cause of the dip in RPA overlap area at 4 hours but speculate that it could be due to changing compaction levels of the resected DNA or competition with RAD51 which, for whatever reason, leads to RAD51 being the dominant ssDNA binder at 4 hours (In agreement with this, RAD51 peaks at 4 hours). Importantly, we consider the BrdU/ssDNA signal to be a good measure of resection and although it peaks at 4 hours, it remains significantly above control levels at later time points implying persistent ssDNA (SI Figure 6). We will include this information in the manuscript on line 189:

A small but significant decrease in RPA signal at 4 hours coincides with peak RAD51 colocalization and likely indicates that at this time ssDNA/RAD51

interactions are more prevalent than ssDNA/RPA interactions, however it is also possible that the decrease in RPA signal is due to changes in the compaction or organization of the resected ssDNA/RPA resulting in a smaller overlap area.

(5) The rationale that RAD52 is required for early RAD51 recruitment is unclear. For consolidating this idea, the authors should show no RAD51 recruitment in RAD52/BRCA2 double-knockdown cells. Because double-knockdown is lethal, the experiment may be infeasible. However, since RAD52/BRCA1 double-knockdown was examined, it is worthwhile to test RAD52/BRCA2 knockdown as well. In the manuscript, the authors state that ‘Combined RAD52/BRCA1 deficiencies have been shown to lead to synthetic lethality similar to that seen in combined RAD52/BRCA2 deficient cells’. Thus, testing RAD52/BRCA2 knockdown at earlier time points may be feasible. If the experiment is infeasible, the authors should explicitly state why the results support the conclusion that RAD52 is required for RAD51 recruitment at an early phase, and is not sustained at ssDNA in the absence of BRCA2.

Like reviewer 1, reviewer 2 raises a good point. (Copied from response to reviewer 1): The reviewer raises a good point and we have spent significant time over the past few months attempting to optimize and image double knock-down siRAD52/siBRCA2 cells. As Reviewer 2 predicted, the combined synthetic lethality of these two deficiencies prohibits any meaningful imaging, as siRNA treatment resulted in rapid cell death. Similar, although slightly less severe results, were seen in the siRAD52/siBRCA1 cells which could not be imaged for more than a couple of hours after CPT treatment and also showed a much lower percentage of cells in S-phase (based on EdU label). While we could not measure RAD51 recruitment in siRAD52/siBRCA2 cells, we could confirm the synthetic lethality of these deficiencies which agrees well with our conclusions. We have added the following to line 315 to address this:

“Attempts to prepare double-depleted siRAD52/siBRCA2 cells similarly demonstrated this synthetic lethality with most cells succumbing to apoptosis during the transfection process.”

6. *The role of BRCA1 for RAD51 filament formation by the recruitment of BRCA2 via PALB2 has been demonstrated in several studies (Buisson et al., 2017; Orthwein et al., 2015; Zhang et al., 2009). Therefore, a major finding in Figures 5 and 6 is that RAD52 promotes BRCA2 in the absence of BRCA1. In Figure 5E, the authors demonstrate a reduction in BRCA2 recruitment in BRCA1-depleted cells, but importantly, BRCA2 recruitment is not completely abolished. In contrast to results for BRCA1 knockdown alone, the authors show no BRCA2 recruitment in RAD52/BRCA1 double-knockdown cells. Even though this is an important finding, the authors unfortunately used different time points. The authors should test BRCA2 recruitment in control, BRCA1, RAD52 and RAD52/BRCA1 double-knockdown cells at the same time point. If RAD52 has a role in promoting BRCA2, an arrow from RAD52 to BRCA2 should be shown in Figure 6C (ii) for WT. This will provide novel insights into the mechanism of RAD51 loading via BRCA1/2 in response to replication stress.*

The reviewer offers an alternate explanation to that which we offer in the manuscript and while we find it interesting, we maintain that the data better support our model. Whereas the reviewer suspects RAD52 is mediating BRCA2 dependent ssDNA/RAD51 association, we believe RAD52 mediates this association directly. Our primary evidence for this is twofold: firstly the lack of BRCA2 observed at early ssDNA/RAD51 association in both siRAD52 and siBRCA1 cells (now incorporated into Figure 6E at 2 hours), and secondly the lack of difference in any examined ssDNA/RAD51 association at early time points in siBRCA2 cells (Figure 5A). If RAD52 was capable of mediating BRCA2 but not of orchestrating ssDNA/RAD51 association, then we would expect a much more severe abrogation of repair in siBRCA2 cells during early HR – instead we detect delays in resection. Furthermore, the reviewer's model does not explain the synthetic lethality we have now observed for siRAD52/siBRCA2. Unfortunately, due to the enhanced cell death in these treated cells, repeated endeavours to image cells later than two hours after CPT treatment in siRAD52/siBRCA1 cells also precludes comparison of BRCA2 recruitment at the later time points shown in Figure 6E. We have, however, reconfigured Figures 5B and 6E so that WT, siBRCA1 and siRAD52 levels of BRCA2 can better be visualized together, as the reviewer recommended.

Currently we do not have a conclusive explanation for the small amount of later BRCA2 recruitment observed in siBRCA1 cells, potentially it is due to insufficient

knockdown of BRCA1 or PALB2 independent association of BRCA2 with RAD51, as reported in vitro (Jensen et al, Nature). However we do maintain that our model of RAD52 mediation of ssDNA/RAD51 association independent of BRCA2 is best supported by the data. We do not believe incorporation of the reviewer's alternate model would benefit our manuscript, although we thank them for the thought-provoking idea.

Minor points:

In multi-colour staining, each colour may show a one- or half-pixel shift. The authors should provide information regarding pixel shift for each colour. Pixel shift should be examined using multi-colour TetraSpeck™ beads.

This information is included in the methods sections. We note that all chromatic aberration in our imaging systems are corrected using a TetraSpeck mapping sample, which is integrated in our single-molecule mapping routine, as detailed in previous work (Reid, DA. et al. PNAS, 2015, 10.1073/pnas.1420115112; Yin, Y.D & Rothenberg, E., Sci. Rep., 2016, 10.1038/srep30819; Bermudez-Hernandez, K. et al., Sci. Rep., 2017, 10.1038/s41598-017-14922-8) . Specifically, we work with effective ~100 nm pixels but render in super-resolution at 10-20 nm. Any 100 nm or more shift would be very apparent in the data. To avoid this, we conduct chromatic mapping at the beginning of each experiment (usually once every 3-6 hours) and apply these maps to either our input raw TIFF stacks, or to our output super-resolution images. We detail this in our methods, line 873:

To correct for camera offsets and chromatic aberrations caused by the varying diffraction behaviors of different energy emissions a polynomial morph-type mapping algorithm was applied to the three-color images. The correcting map was generated before each experiment by imaging spatially separated Tetraspecks in each of the three channels (Tetraspecks, 100 nm, Life Technologies, T7279). The localizations of the beads were calculated by independently fitting their diffraction-limited Point Spread Functions (PSF) with Gaussian functions. The localizations could then be matched across the three color channels and an elastic mapping matrix built from the polynomial morph-type mapping function using an IDL (Exelis Visual Information Solutions) custom script. Raw multicolor cell image stacks were then spatially corrected before SR rendering and analysis.

We have also specifically quantified the mapping error and incorporated that as SI Figure 7 and in the Methods:

“Mapping error distances were calculated by imaging multicolored beads and found to be < 10 nm when mapping the blue channel to the red, and ~ 10 nm when mapping the green channel to the red (SI Figure 7).”

8. In Figure 3A, the purpose of the small panels is unclear, although the authors state that they are representative images. For example, in the middle panel of Figure 3A, MRE11 foci do not overlap with naDNA. The authors should clarify the intended meaning of this point.

This highlights some of the difficulties in presenting super resolution renderings as images in publications as we now discuss in the SI discussion. These foci all represent foci with overlaps, however the pixel brightening to make them visible in print does not ensure clear overlap. We will clarify this point in the Figure captions showing SR images by adding:

“Super-resolved images have been brightened and smoothed to better show clustered and overlapping signals.”

Reviewer #3 (Remarks to the Author):

Rothenberg and coworkers use Super-resolution fluorescence imaging to observe the temporal dynamics of 4 proteins associated with DNA repair through homologous recombination. They use colocalization of different protein clusters/ puncta on the damage site to establish the order of arrival, cross talk and epistasis between the proteins BRCA1 and 2 and RAD51 and 52. While this study is very informative for the field of DNA damage and repair, this reviewer does not see any novelty in the approach.

We thank the reviewer for acknowledging how informative our research is regarding the HR process and although we disagree with their point regarding the lack of novelty in our methodology, the new insights into DNA damage and repair are,

indeed, the primary point of this publication. With regards to the methodology, as the previous reviewer pointed out, this is one of the first large-scale applications of multi-color single molecule super resolution imaging applied to the complex DDR process. Moreover, the assays we use here, both in terms of sample preparation and analysis, are predominantly new and will be required, moving forward, for many other DDR research projects.

Further, the study relies on STORM imaging, yet there is no information on critical SR imaging aspects such as localization precision and most importantly channel dependent and field dependent registration error have been completely ignored. There is no information on background fluorescence, # of localizations per cluster etc.

A key aspect of our research – and one that controls for the vast majority of imaginable imaging artifacts, unlike in other SR studies – is our analytical methodology which inherently normalizes our data to remove the errors discussed by the reviewer. Moreover, these values are not generally reported in applied SR manuscripts as it is understood that they are generally unbiased errors and therefore unlikely to result in false positive data. A new discussion of this methodology and its advantages is now included in the SI Discussion:

“ With hundreds to thousands of individual protein signals (detected a clustered fluorophore localizations attributable to either single proteins or multiple copies) detected in each cell, the rendered pseudo-images did not lend themselves to straightforward overlap quantification. Thus, in order to quantify association between naDNA, proteins, and γ H2A.X, we used a Monte Carlo simulation to randomly redistribute clusters of one color within the nuclear region of interest while maintaining the ‘real’ distribution of the second color. This allowed us to calculate the number, area, and percentage of overlaps expected from a randomly associated pair of species specific to each cell imaged. This normalized each simulation to both area and to cell-to-cell differences in naDNA and protein density. To generate a colocalization coefficient which described the association relative to random, the number or area of overlaps in the real image was ratioed to an average taken from twenty simulations of the same cell. In this context, a colocalization coefficient of 2 showed a doubling in the number or area of overlaps above random, and 3 showed

a tripling. Colocalization coefficients calculated for undamaged control cells were routinely determined to be above random due to HR protein associations with endogenously stressed and broken RFs, and also because of their transient binding to unwound DNA. For example in Figure 2A, control levels of RAD51 overlap with naDNA were significantly higher than random (Colocalization of ~1.5 compared to 1 when random) while RAD52 was slightly higher. Accordingly, we determined changes in colocalization comparative to control levels – for example, after 1 hour of recovery, both RAD51 and RAD52 were detected at control levels, before significantly accumulating above control levels in the following hour of repair and remained prevalently colocalized with the naDNA foci until more than 12 hours of recovery (Figure 2A-B). In only a handful of cases were specific proteins detected as actively excluded from undamaged replication foci (for example BRCA2, Figure 2C).

Controls levels are thus crucial in assessing changes in associations, whereas the randomized simulation approach is more important in normalizing for differences in cell size, cluster size, cluster number, etc. Effectively, we are normalizing each cell's overlaps to its own eccentricities, allowing for sensitive detection of changes across the cell population upon DDR. We have previously published an extensive justification and discussion of this method, as well as the associated calculations and ImageJ plugin.”

We have also endeavored to address the reviewer's specific requests and have incorporated mapping and localization errors into SI Figure 7 as follows:

“Calculation of mapping and localization errors.

(A-B) To evaluate the mapping errors, we imaged Tetraspeck beads (> 300 beads for one experimental set) in 3 color channels (647, 568, and 488 nm channels), and transformed the beads positions in 488 and 568 channels using the same polynomial functions generated during channel alignment (see Methods), respectively. The Mapping Error was then evaluated by calculating the distance from the transformed beads position to their reference positions in 647 channels. Usually, the mapping error from the 488 channel to the 647 (ref channel) is < 10 nm while the mapping error from 568 channel to the 647 (ref channel) is ~ 10 nm.

(C) The fitting uncertainty (std) was estimated following the formula given in the figure, where s is the fitted sigma of a Gaussian modeled PSF (we use half of the

FWHM given by QuickPalm, which is a little bigger than the sigma), and a is the dimension of the pixel size; b stands for the camera background and N denotes the photon number of each blinking event. The maximum of the uncertainty distribution peaks at ~16, 21, 31 nm for the 647, 488, and 568 channels, respectively. The distribution can be fitted into an exponential modified Gaussian distribution, resulting in centers of such Gaussian distributions a bit left-shift from their maximum locations (~9.5, 9.3, 17.7 nm for CH647, CH488, and CH568, respectively).”

And into the methods section, line 883:

“Mapping error distances were calculated by imaging multicolored beads and found to be < 10 nm when mapping the blue channel to the red, and ~ 10 nm when mapping the green channel to the red (SI Figure 7). Fitting uncertainty was also determined to be ~9.5, 9.3, 17.7 nm in red, blue, and green, respectively (SI Figure 7).”

Background fluorescence does not impact SR imaging beyond limiting the fitting uncertainty and our values are generally low. We have included this information in the methods section as well, at line 887:

“The camera reading background varies with different microscopy configurations. In brief, with our current illumination and EM gain configurations (as detailed in the method section), the camera reading background is ~ 230, ~170, and ~320 analog-to-digital unit (ADU) in AF647, AF568, and AF488 channel, respectively. Such background as well as other background signal (e.g. out-of-focus fluorescence, auto-fluorescence) is subtracted before localizing individual single-molecule. Besides the already subtracted camera reading background, the background for single-molecule localization is characterized by calculating the standard deviation on a small area of 2×2 micron² centered on the minimum intensity pixel of each frame of the image stack, and such background is usually about ~12 - 16 ADU.”

Clusters were assessed by automated thresholding of the images, thus removing single fluorophore and aberrant signals from further analysis. However, because this is a single-molecule based approach, single proteins detected using an antibody with multiple fluorophores conjugated could still be detected and included in the analysis – this was the particular strength of our assays, as is now discussed in the new SI Discussion. Furthermore, by not placing a lower limit on cluster

requirements, we were able to reach a level of detection that confocal measurements cannot, ie. Beyond the dozens of colocalized fluorophores usually required to discern a 'focus'.

It is clear that any claims based on co-localization of fluorescence spots in multiple channels must be backed by rigorous controls, both biological and through optics/ simulations.

We are confident that our analytical approach and controls are adequate and we detail our reasoning in our added SI Discussion. Specifically, we have quantified two control levels. Firstly, the Monte Carlo simulations provided normalization of every image to the predicted random level of overlap within that same image. This accounted for the high density of biomolecules within the nucleus and the non-zero overlap count caused by these completely random colocalizations, especially when projected 3D information into a 2D image. Secondly, we determined colocalization between proteins and naDNA in undamaged, S-phase cells. In almost all circumstances, this was found to be above a random amount and we determined that this was due mostly to transient or off-target interactions between these molecules in the biological environment. This was especially supported by the high degree of RPA/naDNA overlap in control cells due to RPA association at undamaged RFs. Calculation of control levels of overlap also takes into account cross-talk and bleed-through and revealed very minimal influence from these artifacts, as demonstrated by the near-random control levels seen for Ku and BLM (SI Figure 6). For further evidence that use of the randomized Monte Carlo simulation approach adequately controls for bias please see our recent methods paper: Bermudez-Hernandez, et al. Sci Rep, 14882, 2017.

This reviewer finds the manuscript lacking these crucial pieces of information. Based on these observations, this reviewer does not recommend the publication of this manuscript in Nat. Comm. in its current form and further recommends that the authors target their manuscript to a more specialized journal.

We are disappointed that this is the stance the reviewer chooses to take, particularly as they have chosen to dismiss that actual content of the paper purely because some highly specialized details were not included in the manuscript. Had the

reviewer asked for these numbers and then taken the time to assess the biological logic of the paper with the assumption that these numbers would be provided, we would hope that they would have seen its worth. Their own assertion that our study is “very informative for the field of DNA damage and repair” seems at odds with their recommendation not to publish. Furthermore, we stress that for those working within DNA damage and repair, this study presents an exciting advance in technique and opens up many new possible avenues for study. This is highlighted by reviewer 2’s acknowledgement that “this is the first study that demonstrates the spatiotemporal dynamics in HR proteins using super-resolution microscopy.” Having now provided the requested technical detail, and remaining open to further control experiments if the reviewer can detail and justify them, we hope they will reconsider their opinion of our work.

REVIEWERS' COMMENTS:

Reviewer #1 (Remarks to the Author):

The authors satisfactorily addressed my points.

Reviewer #2 (Remarks to the Author):

The authors have adequately addressed my concerns. Although I feel the relationship between BRCA1/2, RAD52 and RAD51 in this pathway should be consolidated by other molecular techniques in future, this study has clearly proven that super-resolution microscopy is a powerful tool and provides us many information in the context of protein recruitment at DNA damage site. Thus, in my opinion, this interesting work is suitable for publication.

Reviewer #3 (Remarks to the Author):

This reviewer thanks the authors for their time and effort in addressing the comments raised. As the authors, who have developed their SR imaging and analyses pipeline may already know, interpretation of SR results is highly sensitive to three things: labeling methodology, optics and analyses of the data. While the focus of the current study is the biological finding, it should not be devoid of necessary details that will help in establishing the accuracy of the interpretations as well as reproducibility by others. This reviewer thanks the authors for including the details in the manuscript now. It will definitely help with reproducibility of the experiments thus enabling their reach to a wider audience.

This reviewer has two main reservations with the manuscript and still questions the novelty and the accuracy of the biological findings:

1. In the light of the claims made by the authors about the importance of their findings, this reviewer believes that new findings in biology must be backed by at least a couple of different assays. In this case, while the SR imaging and analyses pipeline are pretty robust (as ascertained from the new details on SR imaging), this reviewer is aware of artefacts due to antibody labeling and thus takes these results with a grain of salt. The reviewer encourages the authors to test the sequence of arrival of at least a few proteins (or at the least a couple of cases) using alternative tagging methods, where possible. Additionally, (or alternatively), biochemical analyses of protein levels (using western blot/ pull down) that support some of the findings (such as the finding that early BRCA2 activity is not detected due to transient interactions) are a must.

2. Since the studies relies so heavily on antibody labeling, one concern that must be addressed is the relative sensitivities of these antibodies. It seems that this point has been raised by another reviewer as well, however, the authors have chosen to ignore addressing this point.

In the light of these reservations, this reviewer does not recommend publication of the manuscript in Nature communications.

Point-by-point response to the reviewers' comments for manuscript No. NCOMMS-18-01634-A

Spatiotemporal Dynamics of Homologous Recombination Repair at Single Collapsed Replication Forks

Donna R. Whelan, Wei Ting C. Lee, Yandong Yin, Dylan M. Ofri, Keria Bermudez-Hernandez, Sarah Keegan, David Fenyo & Eli Rothenberg.

Black text denotes reviewer comments. Blue text denotes point-by-point responses. Red denotes changes to the manuscript and SI.

REVIEWERS' COMMENTS:

Reviewer #1 (Remarks to the Author):

The authours satisfactorily addressed my points.

We thank this reviewer for his/her support of our manuscript.

Reviewer #2 (Remarks to the Author):

The authors have adequately addressed my concerns. Although I feel the relationship between BRCA1/2, RAD52 and RAD51 in this pathway should be consolidated by other molecular techniques in future, this study has clearly proven that super-resolution microscopy is a powerful tool and provides us many information in the context of protein recruitment at DNA damage site. Thus, in my opinion, this interesting work is suitable for publication.

We appreciate this reviewer's enthusiasm from our study and support for our manuscript. To address the remark regarding the use of other techniques to further study this pathway the following was added to our discussion section:

These findings will form the basis for future work incorporating other molecular techniques to further reveal the *in vivo* nature of BRCA1, BRCA2, RAD51, and RAD52 and their crosstalk.

Reviewer #3 (Remarks to the Author):

This reviewer thanks the authors for their time and effort in addressing the comments raised. As the authors, who have developed their SR imaging and analyses pipeline may already know, interpretation of SR results is highly sensitive to three things: labeling methodology, optics and analyses of the data. While the focus of the current study is the biological finding, it should not be devoid of necessary details that will help in establishing the accuracy of the interpretations as well as reproducibility by others. This reviewer thanks the authors for including the details in the manuscript now. It will definitely help with reproducibility of the experiments thus enabling their reach to a wider audience.

This reviewer has two main reservations with the manuscript and still questions the novelty and the accuracy of the biological findings:

1. In the light of the claims made by the authors about the importance of their findings, this reviewer believes that new findings in biology must be backed by at least a couple of different assays. In this case, while the SR imaging and analyses pipeline are pretty robust (as ascertained from the new details on SR imaging), this reviewer is aware of artefacts due to antibody labeling and thus takes these results with a grain of salt. The reviewer encourages the authors to test the sequence of arrival of at least a few proteins (or at the least a couple of cases) using alternative tagging methods, where possible. Additionally, (or alternatively), biochemical analyses of protein levels (using western blot/ pull down) that support some of the findings (such as the finding that early BRCA2 activity is not detected due to transient interactions) are a must.

We disagree with this comment. The molecular biology methods proposed by this reviewer, such as Western-Blot, rely on the use of antibodies and will be subjected to

similar limitations as immuno-fluorescence. Moreover, WB and similar biochemical assays cannot offer detection comparable to the sensitivity provided by single-molecule imaging, nor are these methods useful for detecting singular events and sub-populations within individual cells as detailed in our analysis pipeline that was uniquely developed for our present studies. Protein tagging methods are also subject to limitations and artifacts, which can strongly affect the recruitment kinetics and outcome of such studies (see for instance Kochan et al, NAR 2017). While we appreciate the inherent fallibility of all science and are acutely aware of the limitations of our studies we do believe we have made important and, as you say, robust deductions regarding an exceptionally important biological pathway which should be published in Nature Communications. We anticipate, and welcome, future studies testing and building on our work presented here.

2. Since the studies relies so heavily on antibody labeling, one concern that must be addressed is the relative sensitivities of these antibodies. It seems that this point has been raised by another reviewer as well, however, the authors have chosen to ignore addressing this point.

We note that many biochemistry studies rely on antibody labelling and that, although an imperfect approach, it remains a cornerstone technique because of its exceptional usefulness. Moreover, the limitations of antibody labelling and interpretation of immunofluorescence data are well documented within the field and were extensively considered during the planning of our experiments and analyses. To address the reviewer's comment and demonstrate the minimal impact even large antibody sensitivity differences could have on our findings, we have included a new paragraph in our detailed supplementary discussion of our study's limitations:

A further potential concern we considered was differences in antibody sensitivity causing a bias in images towards detection of the more sensitively detected proteins. For this reason, our analyses focused primarily on assays that would not be affected by sensitivity differences; both the quantification of protein overlap with naDNA foci and the protein-protein association distribution avoid this problem. In the former case, the level of overlap of a particular protein is normalized to control levels of overlap and to random simulations, both of which carry the same inherent antibody

sensitivity and therefore, with normalization, remove this potential confounding error. In the case of the spatial analysis both target proteins must be labelled in order for the analysis to be carried out so there is no potential bias here to a more or less sensitive antibody. An error would only arise in this analysis if an entirely insensitive (absent or non-specific) antibody was used, something we avoided by using antibodies that had already been validated prior to use. The intrafoci analysis (Figures 2D, 3A, 3C, 3D) is susceptible to bias from differences in antibody sensitivity and for this reason we limited ourselves to qualitative conclusions drawn from these analyses that could be strengthened by other observations. With more specific approaches to labeling, we envision an increase in sensitivity allowing for smaller scale changes in the accumulation of proteins such as RAD51 and RPA to be monitored, as well as quantitative intrafoci analysis. Furthermore, stoichiometric labeling will allow quantification of HR proteins at these single DSBs^{20,21}.

In the light of these reservations, this reviewer does not recommend publication of the manuscript in Nature communications.